# Cut Your Losses
# in Large-Vocabulary Language Models

**Erik Wijmans**[*] **Brody Huval** **Alexander Hertzberg** **Vladlen Koltun** **Philipp Krähenbühl**
Apple

## Abstract

As language models grow ever larger, so do their vocabularies. This has shifted the memory footprint of LLMs during training disproportionately to one single layer: the cross-entropy in the loss computation. Cross-entropy builds up a logit matrix with entries for each pair of input tokens and vocabulary items and, for small models, consumes an order of magnitude more memory than the rest of the LLM combined. We propose Cut Cross-Entropy (CCE), a method that computes the cross-entropy loss without materializing the logits for all tokens into global memory. Rather, CCE only computes the logit for the correct token and evaluates the log-sum-exp over all logits on the fly. We implement a custom kernel that performs the matrix multiplications and the log-sum-exp reduction over the vocabulary in flash memory, making global memory consumption for the cross-entropy computation negligible. This has a dramatic effect. Taking the Gemma 2 (2B) model as an example, CCE reduces the memory footprint of the loss computation from 24 GB to 1 MB, and the total training-time memory consumption of the classifier head from 28 GB to 1 GB. To improve the throughput of CCE, we leverage the inherent sparsity of softmax and propose to skip elements of the gradient computation that have a negligible (i.e., below numerical precision) contribution to the gradient. Experiments demonstrate that the dramatic reduction in memory consumption is accomplished without sacrificing training speed or convergence.

https://github.com/apple/ml-cross-entropy

## 1 Introduction

Progress in large language models (LLMs) has been fueled in part by an increase in parameter count, context length, and vocabulary size (the number of tokens that can be used to represent the input). As LLMs grew, so did the associated infrastructure. Large mini-batch gradient descent (Goyal et al., 2017) combined with data-parallelism (Hillis & Steele, 1986) enabled the harnessing of increasing computational power. ZeRO (Rajbhandari et al., 2020) broke the dependence between the number of GPUs and the memory used for model parameters, gradients, and optimizer state. Activation checkpointing (Chen et al., 2016) reduced the amount of memory used for activations, supporting the development of deeper models. FlashAttention (Dao et al., 2022) reduced the memory used in self-attention from $O(N^2)$ to $O(N)$, thereby supporting longer context windows. These improvements gradually shifted the memory consumption of LLM training to one single layer – the cross-entropy loss, whose memory footprint grows with the product of vocabulary size and number of tokens per batch. The cross-entropy loss is responsible for up to 90% of the memory footprint of modern LLM training (see Fig. 1a). The problem grows only more acute with time, since even the largest contemporary vocabularies (e.g., 256K tokens) may benefit from further expansion (Tao et al., 2024).

We propose a cross-entropy implementation, Cut Cross-Entropy (CCE), that has a negligible memory footprint and scales to arbitrarily large vocabularies. Our key insight is that computation of the loss and its gradient only depends on a single log-probability, that of the ground-truth label. With an arithmetic reformulation, we decompose the cross-entropy loss into an index matrix multiplication over a single ground-truth label and a log-sum-exp operation over all vocabulary entries for each token. Each operation has small and well-defined inputs – the network embeddings and classifier

---

[*]Corresponding author: ewijmans@apple.com

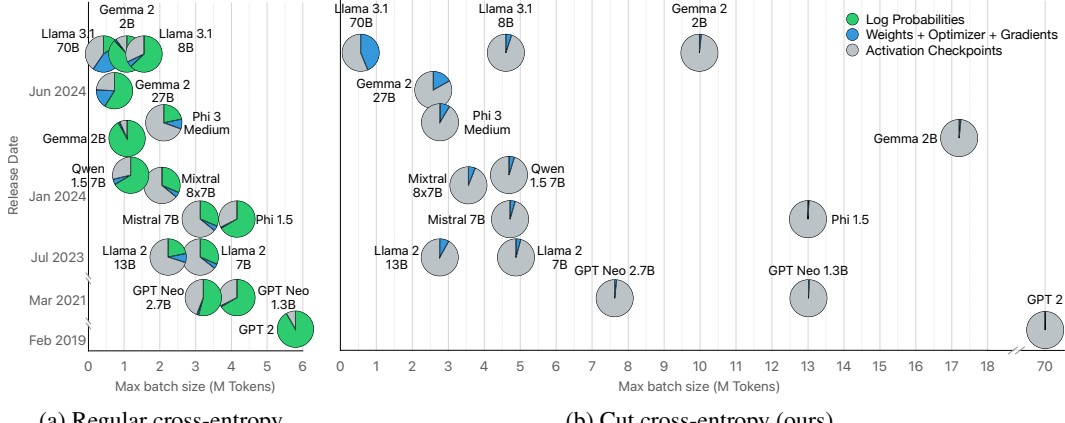

(a) Regular cross-entropy          (b) Cut cross-entropy (ours)

Figure 1: Memory use and maximum attainable batch size (in millions of tokens) for a variety of frontier models on a 16-GPU (80 GB each) fully-sharded data-parallel setup (Rajbhandari et al., 2020) with activation checkpointing (Chen et al., 2016) and a mixed-precision 16-bit (fp16/bf16) AdamW optimizer (Kingma & Ba, 2015; Loshchilov & Hutter, 2019). For each model, we break its memory use down into weights and optimizer states, activation checkpoints, and the log-probabilities computed by the cross-entropy loss layer. Our Cut Cross-Entropy (CCE) enables increasing the batch size by 1.5x (Llama 2 13B) to 10x (GPT 2, Gemma 2 2B), with no sacrifice in speed or convergence. Exact values in Table A4.

matrix – and a single scalar output per token. Both operations do, however, rely on a large intermediate logit matrix that computes the score for each token and potential vocabulary entry. We show that there is no need to materialize this logit matrix in GPU memory. Instead, we compute logits as needed in SRAM in a series of custom CUDA kernels. The result is a cross-entropy computation that has negligible memory footprint, with no detrimental effect on latency or convergence. See Fig. 1b for a breakdown of memory savings and consequent batch size increases afforded by CCE.

## 2   RELATED WORK

**Attention mechanisms.** The effectiveness of transformers (Vaswani et al., 2017) in modeling language has drawn attention to their compute and memory requirements. Multiple works have proposed alternatives to scaled dot-product attention that reduce transformers' computation and memory (Kitaev et al., 2020; Wang et al., 2020; Choromanski et al., 2021). Other model classes, such as structured state-space models (Gu et al., 2022; Gu & Dao, 2023), have also shown promising results. We study a different part of the model – its classifier head – that is not considered in these works.

**Attention implementations.** In addition to alternative attention mechanisms, the community has also tackled the daunting memory consumption of LLMs via efficient implementations. Rabe & Staats (2021) developed a self-attention implementation that makes use of chunking. Chen et al. (2023) proposed an implementation that broke the operation into two stages, reduction and matrix multiplication. This makes efficient use of GPU memory and registers but requires recomputation in the forward pass. FlashAttention (Dao et al., 2022) uses an online softmax (Milakov & Gimelshein, 2018) and, like CCE, materializes blocks of the $N^2$-sized self-attention matrix in on-chip SRAM rather than slower global DRAM. This is one of the key ideas that CCE builds on to develop a memory-efficient cross-entropy formulation.

**Vocabulary reduction.** One way to minimize the amount of memory used by the log-probabilities over the tokens is to reduce the number of 'active' tokens in the vocabulary. Grave et al. (2017) proposed to use a vocabulary with a hierarchical structure, thereby requiring the log-probabilities for only a subset of the vocabulary at any given time. Yu et al. (2023) explore tokenization-free byte-level models that operate on dramatically smaller vocabularies.

**Sequence and model parallelism.** Sequence parallelism (Jacobs et al., 2023; Li et al., 2023) enables training very large models (with large vocabularies) by splitting an individual input sequence across

multiple GPUs. Various model parallelism techniques (Huang et al., 2019; Narayanan et al., 2019; Shoeybi et al., 2019) achieve the same goal of training very large models (with large vocabularies) by distributing the computation and memory consumption of different pieces across multiple GPUs.

**Efficient cross-entropy implementations.** A number of recent implementations use chunking to reduce the memory usage of the cross-entropy layer. Yet chunking induces a trade-off. Memory footprint is minimized when the number of chunks is high, but latency is minimized when the number of chunks is low. CCE utilizes only on-chip SRAM and minimizes both memory footprint and latency. Liger Kernels (Hsu et al., 2024) make efficient use of the GPU via chunking and by computing the loss+gradient simultaneously. The latter requires that any transform applied to the loss (such as masking) is implemented in the kernel itself. CCE has separate forward and backward stages, enabling user-defined transformations on the loss.

## 3 PRELIMINARIES

Let $P(x) = \prod_{i=1}^{N} P(x_i \mid x_1 \ldots x_{i-1})$ be a Large Language Model (LLM) over a vocabulary $V$. The LLM parameterizes an autoregressive distribution over all possible tokens $x_i \in V$ given the preceding $N - 1$ tokens. Specifically, this distribution is the combination of a backbone network $f : x_1 \ldots x_{i-1} \to \mathbb{R}^D$ and a linear classifier $\mathbf{C} \in \mathbb{R}^{D \times |V|}$:

$$P(x_i \mid x_1 \ldots x_{i-1}) = \text{softmax}_{x_i}(\mathbf{C}^\top f(x_1 \ldots x_{i-1})), \tag{1}$$

$$\text{softmax}_k(\mathbf{v}) = \frac{\exp(v_k)}{\sum_j \exp(v_j)}. \tag{2}$$

The backbone network $f(x_1, \ldots, x_{i-1}) \in \mathbb{R}^D$ encodes a token sequence in the $D$-dimensional feature vector. The linear classifier $\mathbf{C} \in \mathbb{R}^{D \times |V|}$ projects the embedding into an output space of the vocabulary $V$. The $\text{softmax}_k(\mathbf{v})$ produces the probability over all vocabulary entries from the unnormalized log probabilities (logits) produced by $\mathbf{C}^\top f(x_1 \ldots x_{i-1})$.

### 3.1 VOCABULARY

LLMs represent their input (and output) as a set of tokens in a vocabulary $V$. The vocabulary is typically constructed by a method such as Byte Pair Encoding (BPE) (Gage, 1994). BPE initializes the vocabulary with all valid byte sequences from a standard text encoding, such as utf-8. Then, over a large corpus of text, BPE finds the most frequent pair of tokens and creates a new token that represents this pair. This continues iteratively until the maximum number of tokens is reached.

Large vocabularies enable a single token to represent multiple characters. This reduces the length of both input and output sequences, compresses larger and more diverse documents into shorter context windows, thus improving the model's comprehension while reducing computational demands.

### 3.2 INFERENCE AND TRAINING

Even with a large vocabulary, sampling from an LLM is memory-efficient at inference time. Specifically, the LLM produces one token at a time, computing $P(x_i | x_1 \ldots x_{i-1})$ and sampling from this distribution (Kwon et al., 2023). Because the distribution over the vocabulary is only needed for a single token at a time, the memory footprint is independent of sequence length.

At training time, the LLM maximizes the log-likelihood of the next token:

$$\ell(\hat{\mathbf{x}}) = \sum_{i=1}^{N} \log P(\hat{x}_i | \hat{x}_1, \ldots, \hat{x}_{i-1}). \tag{3}$$

Due to the structure of most backbones (Vaswani et al., 2017; Gu et al., 2022; Gu & Dao, 2023), $f(x_1), f(x_1, x_2), \ldots, f(x_1, \ldots, x_N)$ is efficiently computed in parallel. However, activations for non-linear layers have to be saved for the backward pass, consuming significant memory. Most LLM training frameworks make use of aggressive activation checkpointing (Chen et al., 2016), sharding (Rajbhandari et al., 2020), and specialized attention implementations (Dao et al., 2022) to keep this memory footprint manageable.

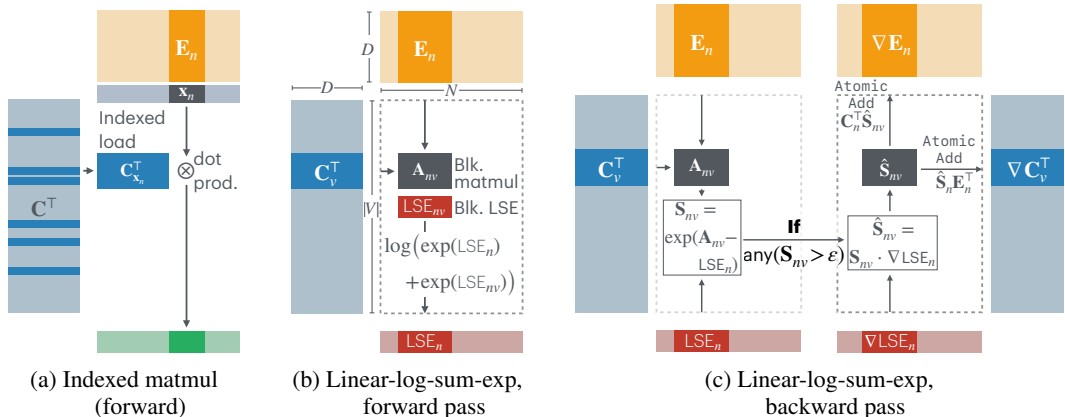

Figure 2: Access patterns and computation of blockwise (a) indexed matrix multiplication, (b) linear-log-sum-exp forward pass, and (c) linear-log-sum-exp backward pass. See Algorithms 1 to 3 for the corresponding algorithms.

With the aforementioned optimizations, the final (cross-entropy loss) layer of the LLM becomes by far the biggest memory hog. For large vocabularies, the final cross-entropy layer accounts for the majority of the model's memory footprint at training time (Fig. 1a). For example, the log-probabilities materialized by the cross-entropy layer account for $40\%$ of the memory consumption of Phi 3.5 (Mini) (Abdin et al., 2024) ($|V| = 32{,}064$), $65\%$ of the memory consumption of Llama 3 (8B) (Dubey et al., 2024) ($|V| = 128{,}000$), and $89\%$ of the memory consumption of Gemma 2 (2B) (Rivière et al., 2024) ($|V| = 256{,}128$). In fact, the log-probabilities of Gemma 2 (2B) for a single sequence $\mathbf{x}$ with length $N = 80{,}000$ use the entire available memory of an $80\,\mathrm{GB}$ H100 GPU. (The sequence length is a factor due to the use of teacher forcing for parallelism.)

We show that a reformulation of the training objective leads to an implementation that has negligible memory consumption above what is required to store the loss and the gradient.

## 4 CUT CROSS-ENTROPY

Consider the cross-entropy loss $\ell_i$ over a single prediction of the next token $P(x_i|x_1 \ldots x_{i-1})$:

$$\ell_i(\mathbf{x}) = \log \operatorname{softmax}_{x_i} \left( \mathbf{C}^\top E_i \right) = C_{x_i}^\top E_i - \log \sum_j \exp \left( C_j^\top E_i \right).$$

Here the first term is a vector product over $D$-dimensional embeddings $E_i = f(x_1 \ldots x_{i-1})$ and a classifier $\mathbf{C}$. The second term is a log-sum-exp operation and is independent of the next token $x_i$. During training, we optimize all next-token predictions $\boldsymbol{\ell} = [\ell_1 \ldots \ell_N]$ jointly using teacher forcing:

$$\boldsymbol{\ell} = \left( \mathbf{C}^\top \mathbf{E} \right)_{\mathbf{x}} - \log \sum_j \exp(C_j^\top \mathbf{E}), \tag{4}$$

where $\mathbf{E} = [E_1 \ldots E_N]$ and $\left( \mathbf{C}^\top \mathbf{E} \right)_{\mathbf{x}} = \left[ C_{x_1}^\top E_1 \ldots C_{x_N}^\top E_N \right]$. The first term in Equation (4) is a combination of an indexing operation and matrix multiplication. It has efficient forward and backward passes, in terms of both compute and memory, as described in Section 4.1. The second term in Equation (4) is a joint log-sum-exp (LSE) and matrix multiplication operation. Section 4.2 describes how to compute the forward pass of this linear-log-sum-exp operation efficiently using a joint matrix multiplication and reduction kernel. Section 4.3 describes how to compute its backward pass efficiently by taking advantage of the sparsity of the gradient over a large vocabulary. Putting all the pieces together yields a memory-efficient low-latency cross-entropy loss.

### 4.1 MEMORY-EFFICIENT INDEXED MATRIX MULTIPLICATION

A naive computation of indexed matrix multiplication involves either explicit computation of the logits $\mathbf{C}^\top \mathbf{E}$ with an $O(N|V|)$ memory cost, or indexing into the classifier $\mathbf{C}_\mathbf{x} = [C_{x_1} \ldots C_{x_N}]$ with

---

**Algorithm 1** Memory-efficient indexed matrix multiplication

---

**Inputs:**      $\mathbf{E} \in \mathbb{R}^{D \times N}$, $\mathbf{C} \in \mathbb{R}^{D \times |V|}$, $\mathbf{x} \in \mathbb{R}^N$.
                 Block sizes $N_B$ and $D_B$.

**Outputs:**  $\mathbf{o} = (\mathbf{C}^\top \mathbf{E})_{\mathbf{x}} \in \mathbb{R}^N$

---

**for** blocks $\mathbf{E}_n, \mathbf{x}_n$ **do**          $\triangleright$ Divide $\mathbf{E}$ and $\mathbf{x}$ into blocks of size $D \times N_B$ and $N_B$, respectively
    $\mathbf{o}_n = \mathbf{0}_{N_B}$                    $\triangleright$ Zero vector of size $N_B$ in on-chip SRAM
    **for** blocks $\mathbf{E}_{n,d}$ **do**             $\triangleright$ Divide $\mathbf{E}_n$ into blocks of size $D_B \times N_B$
        $\mathbf{c} = \mathbf{C}_{\mathbf{x}_n, d}$                   $\triangleright$ Indexed load into on-chip SRAM
        $\mathbf{o}_n \mathrel{+}= \mathbf{E}_{n,d} \cdot \mathbf{c}$                $\triangleright$ Column-wide dot product
    **end for**
    write $\mathbf{o}_n$                          $\triangleright$ From on-chip SRAM to main GPU memory
**end for**

---

an $O(ND)$ memory cost. Our implementation fuses the classifier indexing $\mathbf{C}_{\mathbf{x}}$ with the consecutive dot product between columns $C_{x_i}$ and $E_i$ in a single CUDA/Triton kernel (Tillet et al., 2019). Our kernel retrieves the value $x_i$, the $x_i$-th column from $\mathbf{C}$, and the $i$-th column from $\mathbf{E}$, and stores them in on-chip shared memory (SRAM). It then performs a dot product between $C_{x_i}$ and $E_i$ and writes the result into global memory. The kernel uses only on-chip SRAM throughout and does not allocate any GPU memory. For efficiency, we perform all operations blockwise to make the best use of GPU cache structure. Algorithm 1 and Fig. 2a summarize the computation and access patterns.

## 4.2 MEMORY-EFFICIENT LINEAR-LOG-SUM-EXP, FORWARD PASS

Implementing a serial memory-efficient linear-log-sum-exp is fairly straightforward: use a triple for-loop. The innermost loop computes the dot product between $C_v$ and $E_n$ for the $v$-th token and the $n$-th batch element. The middle loop iterates over the vocabulary, updating the log-sum-exp (LSE) along the way. Finally, the outermost loop iterates over all batch elements. Parallelizing over the outermost loop is trivial and would expose enough work to saturate the CPU due to the number of tokens in training batches (commonly in the thousands). Parallelization that exposes enough work to saturate the GPU is more challenging.

Let us first examine how efficient matrix multiplication between the batch of model output embeddings $\mathbf{E} \in \mathbb{R}^{D \times N}$ and the classifier $\mathbf{C} \in \mathbb{R}^{D \times |V|}$ is implemented on modern GPUs (Kerr et al., 2017). A common method is to first divide the output $\mathbf{O} = \mathbf{C}^\top \mathbf{E} \in \mathbb{R}^{|V| \times N}$ into a set of blocks of size $V_B \times N_B$. Independent CUDA blocks retrieve the corresponding parts $\mathbf{E}_n$ of $\mathbf{E}$ with size $D \times N_B$ and blocks $\mathbf{C}_m$ of $\mathbf{C}$ with size $D \times V_B$, and perform the inner product $\mathbf{O}_{nm} = \mathbf{C}_m^\top \mathbf{E}_n$ along the $D$ dimension. Due to limited on-chip SRAM, most implementations use a for-loop for large values of $D$. They loop over smaller size $D_B \times N_B$ and $D_B \times V_B$ blocks and accumulate $\mathbf{O}_{nv} = \sum_d \mathbf{C}_{vd}^\top \mathbf{E}_{nd}$ in SRAM. Each CUDA block then writes $\mathbf{O}_{nm}$ back into global memory. This method exposes enough work to the GPU and makes efficient use of SRAM and L2 cache.

To produce log-sum-exp($\mathbf{C}^\top \mathbf{E}$), we use the same blocking and parallelization strategy as matrix multiplication. Each block first computes a matrix multiplication, then the log-sum-exp along the vocabulary dimension $m$ for its block, and finally updates LSE with its result.

Note that multiple CUDA blocks are now all writing to the same location of LSE. This includes blocks in the same input range $n$ but different vocabulary ranges $m$. We use a spin-lock on an atomic operation in global memory to synchronize the updates by different CUDA blocks as this is simple to implement in our Triton framework and incurs little overhead. Alternative methods, such as an atomic compare-and-swap loop, may perform better when implementing in CUDA directly.

Algorithm 2 and Fig. 2b summarize the computation and access patterns.

## 4.3 MEMORY-EFFICIENT LINEAR-LOG-SUM-EXP, BACKWARD PASS

The backward pass needs to efficiently compute two gradient updates:

$$\nabla \mathbf{E} = \lambda^\top \frac{\partial}{\partial \mathbf{E}} \log \sum \exp(\mathbf{C}^\top \mathbf{E}) \quad \text{and} \quad \nabla \mathbf{C} = \lambda^\top \frac{\partial}{\partial \mathbf{C}} \log \sum \exp(\mathbf{C}^\top \mathbf{E})$$

---

**Algorithm 2** Memory-efficient linear-log-sum-exp, forward pass

---

| **Inputs:** | $\mathbf{E} \in \mathbb{R}^{D \times N}$ and $\mathbf{C} \in \mathbb{R}^{D \times |V|}$. |
| | Block sizes $N_B$, $V_B$, and $D_B$. |
| **Outputs:** | $\mathrm{LSE} = \log \sum_j \exp(C_j^\top \mathbf{E}) \in \mathbb{R}^N$ |

---

$\mathrm{LSE} = -\infty_N$      $\triangleright$ $-\infty$ vector of size $N$ in main GPU memory
**for** all pairs of blocks $\mathbf{E}_n$, $\mathbf{C}_v$ **do**    $\triangleright$ Divide $\mathbf{E}$ and $\mathbf{C}$ into blocks of size $D \times N_B$ and $D \times V_B$
    $\mathbf{A}_{nv} = \mathbf{0}_{V_B \times N_B}$      $\triangleright$ Zero matrix of size $V_B \times N_B$ in on-chip SRAM
    **for** blocks $\mathbf{E}_{n,d}$, $\mathbf{C}_{v,d}$ **do**    $\triangleright$ Divide $\mathbf{E}_n$ and $\mathbf{C}_v$ into blocks of $D_B \times N_B$ and $D_B \times V_B$
       $\mathbf{A}_{nv} += \mathbf{C}_{v,d}^\top \cdot \mathbf{E}_{n,d}$      $\triangleright$ Blockwise matrix multiplication
    **end for**
    $\mathrm{LSE}_{nv} = \log \sum \exp(\mathbf{A}_{nv}^\top)$      $\triangleright$ Numerically stable implementation with max
    $\mathrm{LSE}_n = \log(\exp(\mathrm{LSE}_n) + \exp(\mathrm{LSE}_{nv}))$    $\triangleright$ Locking thread-safe log-add-exp
**end for**

---

for a backpropagated gradient $\lambda = \nabla \mathrm{LSE}$. Formally, the gradient is defined as

$$\nabla \mathbf{E}^\top = (\mathbf{S} \cdot \nabla \mathrm{LSE})\, \mathbf{C} \quad \text{and} \quad \nabla \mathbf{C}^\top = (\mathbf{S} \cdot \nabla \mathrm{LSE})^\top \mathbf{E}$$

where $\mathbf{S} = \mathrm{softmax}(\mathbf{C}^\top \mathbf{E})$ and $\cdot$ refers to the row-by-row elementwise multiplication of the softmax $\mathbf{S}$ and the gradient $\nabla \mathrm{LSE}$: $\hat{\mathbf{S}} = \mathbf{S} \cdot \nabla \mathrm{LSE}$.

Computationally, the backward pass is a double matrix multiplication $\mathbf{C}^\top \mathbf{E}$ and $\hat{\mathbf{S}}\mathbf{C}$ or $\hat{\mathbf{S}}^\top \mathbf{E}$ with intermediate matrices $\mathbf{S}$ and $\hat{\mathbf{S}}$ that do not fit into GPU memory and undergo a non-linear operation. We take a similar approach to the forward pass, recomputing the matrix $\mathbf{C}^\top \mathbf{E}$ implicitly in the GPU's shared memory. For the backward pass, we do not need to compute the normalization constant of the softmax, since $\mathbf{S} = \mathrm{softmax}(\mathbf{C}^\top \mathbf{E}) = \exp(\mathbf{C}^\top \mathbf{E} - \mathrm{LSE})$. This allows us to reuse the global synchronization of the forward pass, and compute $\mathbf{S}$ efficiently in parallel.

We implement the second matrix multiplication in the main memory of the GPU, as a canonical blockwise implementation would require storing or synchronizing $\mathbf{S}$. Algorithm 3 and Fig. 2c summarize the computation and access patterns. A naive implementation of this algorithm requires zero additional memory but is slow due to repeated global memory load and store operations. We use two techniques to improve the memory access pattern: gradient filtering and vocabulary sorting.

**Gradient filtering.** By definition, the softmax $\mathbf{S}$ sums to one over the vocabulary dimension. If stored in bfloat16 with a 7-bit fraction, any value below $\varepsilon = 2^{-12}$ will likely be ignored due to truncation in the summation or rounding in the normalization.[1] This has profound implications for the softmax matrix $\mathbf{S}$: For any column, at most $\frac{1}{\varepsilon} = 4096$ entries have non-trivial values and contribute to the gradient computation. All other values are either rounded to zero or truncated. In practice, the sparsity of the softmax matrix $\mathbf{S}$ is much higher: empirically, in frontier models we evaluate, less than $0.02\%$ of elements are non-zero. Furthermore, the sparsity of the softmax matrix grows as vocabulary size increases. In Algorithm 3, we take advantage of this sparsity and skip gradient computation for any block whose corresponding softmax matrix $S_{nm}$ has only negligible elements. We chose the threshold $\varepsilon = 2^{-12}$ to be the smallest bfloat16 value that is not truncated. In practice, this leads to a 3.5x speedup without loss of precision in any gradient computation. See Section 5 for a detailed analysis.

The efficiency of gradient filtering is directly related to the block-level sparsity of the softmax matrix. We cannot control the overall sparsity pattern without changing the output. However, we can change the order of the vocabulary to create denser local blocks for more common tokens.

**Vocabulary sorting.** Ideally the vocabulary would be ordered such that all tokens with non-trivial gradients would be contiguously located. This reduces the amount of computation wasted by partially populated blocks – ideally blocks would either be entirely empty (and thus skipped) or entirely populated. We heuristically group the non-trivial gradients by ordering the tokens by their average logit. Specifically, during the forward pass (described in Section 4.2) we compute the average logit

---

[1]The 5 extra bits above the fractional size (7) account for rounding rules, and the consideration that small but not tiny values will likely not get truncated due to the blocking strategies used to compute a sum.

---

**Algorithm 3** Memory-efficient linear-log-sum-exp, backward pass

| | |
|---|---|
| **Inputs:** | $\mathbf{E} \in \mathbb{R}^{D \times N}$, $\mathbf{C} \in \mathbb{R}^{D \times |V|}$, $\mathrm{LSE} \in \mathbb{R}^N$, and $\nabla \mathrm{LSE} \in \mathbb{R}^N$. |
| | Block sizes $N_B$, $V_B$, and $D_B$. |
| | Accuracy threshold $\varepsilon$. |
| **Outputs:** | $\nabla \mathbf{E} \in \mathbb{R}^{D \times N}$, $\nabla \mathbf{C} \in \mathbb{R}^{D \times |V|}$ |

> **for** all pairs of blocks $\mathbf{E}_n$, $\mathbf{C}_v$ **do**      ▷ Divide $\mathbf{E}$ and $\mathbf{C}$ into blocks of size $D \times N_B$ and $D \times V_B$
>     $\mathbf{A}_{nv} = \mathbf{0}_{V_B \times N_B}$      ▷ Zero matrix of size $V_B \times N_B$ in on-chip SRAM
>     **for** blocks $\mathbf{E}_{n,d}$, $\mathbf{C}_{v,d}$ **do**    ▷ Divide $\mathbf{E}_n$ and $\mathbf{C}_v$ into blocks of $D_B \times N_B$ and $D_B \times V_B$
>       $\mathbf{A}_{nv} \mathrel{+}= \mathbf{C}_{v,d}^\top \cdot \mathbf{E}_{n,d}$      ▷ Blockwise matrix multiplication
>     **end for**
>     $\mathbf{S}_{nv} = \exp(\mathbf{A}_{nv} - \mathrm{LSE}_n)$      ▷ Compute the softmax
>     **if** all$(\mathbf{S}_{nv} < \varepsilon)$ **then**
>       **skip**      ▷ Skip computation if below desired numerical precision
>     **end if**
>     **for** blocks $\mathbf{E}_{n,d}$, $\mathbf{C}_{v,d}$ **do**      ▷ Divide $\mathbf{E}_n$ and $\mathbf{C}_m$ into blocks of $D_B \times N_B$ and $D_B \times V_B$
>       $\nabla \mathbf{E}_{n,d}^\top \mathrel{+}= (\mathbf{S}_{nv} \cdot \nabla \mathrm{LSE}_n) \, \mathbf{C}_{v,d}$      ▷ Locking thread-safe gradient update
>       $\nabla \mathbf{C}_{v,d}^\top \mathrel{+}= (\mathbf{S}_{nv} \cdot \nabla \mathrm{LSE}_n)^\top \, \mathbf{E}_{n,d}$      ▷ Locking thread-safe gradient update
>     **end for**
> **end for**

---

per token using an atomic addition. For the backward pass, we divide the vocabulary dimension $|V|$ into blocks with similar average logit instead of arbitrarily. This requires a temporary buffer of size $O(|V|)$, about 1 MB for the largest vocabularies in contemporary LLMs (Rivière et al., 2024).

Putting all the pieces together, we arrive at forward and backward implementations of cross-entropy that have a negligible incremental memory footprint without sacrificing speed. Note that in practice, we found it to be easier and more memory-efficient to merge the indexed matrix-multiplication backward implementation with the backward pass of the linear-log-sum-exp operator (Algorithm 3). The two operations share much of the computation and memory access pattern, see Algorithm 4.

## 5 ANALYSIS

### 5.1 RUNTIME AND MEMORY

First we examine the runtime and memory of various implementations of the cross-entropy loss $\log \mathrm{softmax}_{x_i}(\mathbf{C}^\top \mathbf{E})$. We consider a batch of 8,192 tokens with a vocabulary size of 256,000 and hidden dimension 2,304. This corresponds to Gemma 2 (2B) (Rivière et al., 2024). We use the Alpaca dataset (Taori et al., 2023) for inputs and labels and Gemma 2 (2B) Instruct weights to compute $\mathbf{E}$ and for $\mathbf{C}$. The analysis is summarized in Table 1.

The baseline implements the loss directly in PyTorch (Paszke et al., 2019). This is the default in popular frameworks such as Torch Tune (Torch Tune Team, 2024) and Transformers (Wolf et al., 2019). This method has reasonable throughput but a peak memory usage of 28,000 MB of GPU memory to compute the loss+gradient (Table 1 row 5). Due to memory fragmentation, just computing the loss+gradient for the classifier head requires an 80 GB GPU. torch.compile (Ansel et al., 2024) is able to reduce memory usage by 43% and computation time by 33%, demonstrating the effectiveness of kernel fusion (Table 1 row 4 vs. 5). Torch Tune (Torch Tune Team, 2024) includes a method to compute the cross-entropy loss that divides the computation into chunks and uses torch.compile to save memory. This reduces memory consumption by 65% vs. Baseline and by 40% vs. torch.compile (to 9,631 MB, see Table 1 row 3 vs. 4 and 5). Liger Kernels (Hsu et al., 2024) provide a memory-efficient implementation of the cross-entropy loss that, like Torch Tune, makes uses of chunked computation to reduce peak memory usage. While very effective at reducing the memory footprint, using 95% less memory than Baseline, it has a detrimental effect on latency, more than doubling the wall-clock time for the computation (Table 1, row 2 vs. 4). The memory

---

[2]The gradient and loss are computed simultaneously, not in separate forward/backward passes.

|  | | Loss | | Gradient | | Loss+Gradient | |
|---|---|---|---|---|---|---|---|
|  | Method | Memory | Time | Memory | Time | Memory | Time |
|  | Lower bound | 0.004 MB | | 1,161 MB | | 1,161 MB | |
| 1) | CCE (Ours) | **1 MB** | **46 ms** | **1,163 MB** | 100 ms | **1,164 MB** | 145 ms |
| 2) | Liger Kernels (Hsu et al., 2024)[2] | 1,474 MB | 304 ms | | | 1,474 MB | 304 ms |
| 3) | Torch Tune Team (2024) (8 chunks) | 8,000 MB | 55 ms | 1,630 MB | 115 ms | 9,631 MB | 169 ms |
| 4) | torch.compile | 4,000 MB | 49 ms | 12,000 MB | **92 ms** | 16,000 MB | **143 ms** |
| 5) | Baseline | 24,000 MB | 82 ms | 16,000 MB | 122 ms | 28,000 MB | 208 ms |
| 6) | CCE (No Vocab Sorting) | 0.09 MB | 45 ms | 1,162 MB | 115 ms | 1,162 MB | 159 ms |
| 7) | CCE (No Grad. Filter) | 0.09 MB | 45 ms | 1,163 MB | 314 ms | 1,162 MB | 357 ms |
| 8) | CCE-Kahan | 1 MB | 47 ms | 2,325 MB | 114 ms | 2,326 MB | 160 ms |
| 9) | CCE-Kahan-FullC | 1 MB | 47 ms | 2,326 MB | 268 ms | 2,326 MB | 313 ms |
| 10) | CCE-Kahan-FullE | 1 MB | 47 ms | 2,326 MB | 247 ms | 2,326 MB | 292 ms |

Table 1: Peak memory footprint and time to compute the loss, its gradient, and their combination. Note that intermediate buffers can often (but not always) be reused between the loss and gradient computation, resulting in lower peak memory consumption than the sum of the parts. Batch of $8{,}192$ tokens with a vocabulary size of $256{,}000$ and hidden dimension 2304. Embedding and classifier matrix taken during Gemma 2 (2B) training on Alpaca. Measured on an A100-SXM4 GPU with $80$ GB of RAM, PyTorch 2.4.1, CUDA 12.4, rounded to closest MB. Some numbers are multiples of $1{,}000$ due to dimensions chosen and PyTorch's allocation strategy. 'Lower bound' is the amount of memory required for the output buffer(s), i.e., $\nabla \mathbf{E}$ and $\nabla \mathbf{C}$, this is the lower bound for the memory footprint of any method. Results averaged over 5 seeds.

usage of CCE grows with $O(N+|V|)$, as opposed to $O(N \times |V|)$ for Baseline, torch.compile, and Torch Tune, and $O(N \times D)$ for Liger Kernels. In practice, CCE has a negligible memory footprint regardless of vocabulary size or sequence length.

Compared to the fastest method, torch.compile, CCE computes the loss slightly faster (5%, 4ms, Table 1 row 1 vs. 4). This is because CCE does not write all the logits to global memory. CCE computes the loss+gradient slightly slower (6%, 2 ms). While CCE needs to recompute $\mathbf{C}^\top \mathbf{E}$, it is able to save time in other parts of the computation. See Appendix C.1 for a breakdown of the backwards pass of CCE and Baseline. This increase is largely negligible as the forward+backward pass for even a small LLM (2B parameters) is on the order of seconds.

The performance of CCE is enabled several factors. Without vocabulary sorting CCE takes 15% (23 ms) longer (Table 1 row 1 vs. 6) and without gradient filtering it is 3.4x (356 ms) longer (row 1 vs. 7). CCE utilizes the final gradient floating point type (typically bf16) for summation in global memory. For increased numerical stability, we experiment with Kahan summation (Kahan, 1965) with a higher time and memory cost (Table 1 row 1 vs. 8). We can further incraese the numerical stability by selectively applying gradient filtering to just $\nabla E$ and $\nabla C$. When combined with Kahan summation, removing gradient filtering from either $\nabla C$ or $\nabla E$ results in a similar decrease of performance (Table 1 row 9 or 10 vs. 8). The last variant (CCE-Kahan-FullC) is particularly interesting for pretraining, where the numerical

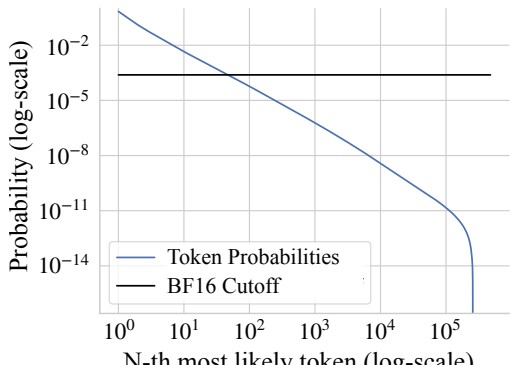

Figure 3: Average probability for the $i$th most likely token, log-log plot. The probabilities very quickly vanish below numerical precision.

precision makes a difference. For fine-tuning all variants of CCE perform equivalently, as shown in Section 5.3.

In Appendix B, we demonstrate that CCE (and other methods) can be made up to 3 times faster by removing tokens that are ignored. In Appendix C we benchmark with more models. We find that as the vocabulary size ($|V|$) to hidden size ($D$) ratio decreases, CCE's advantage in computation time for Loss+Gradient decreases, but continues to save a substantial amount of memory.

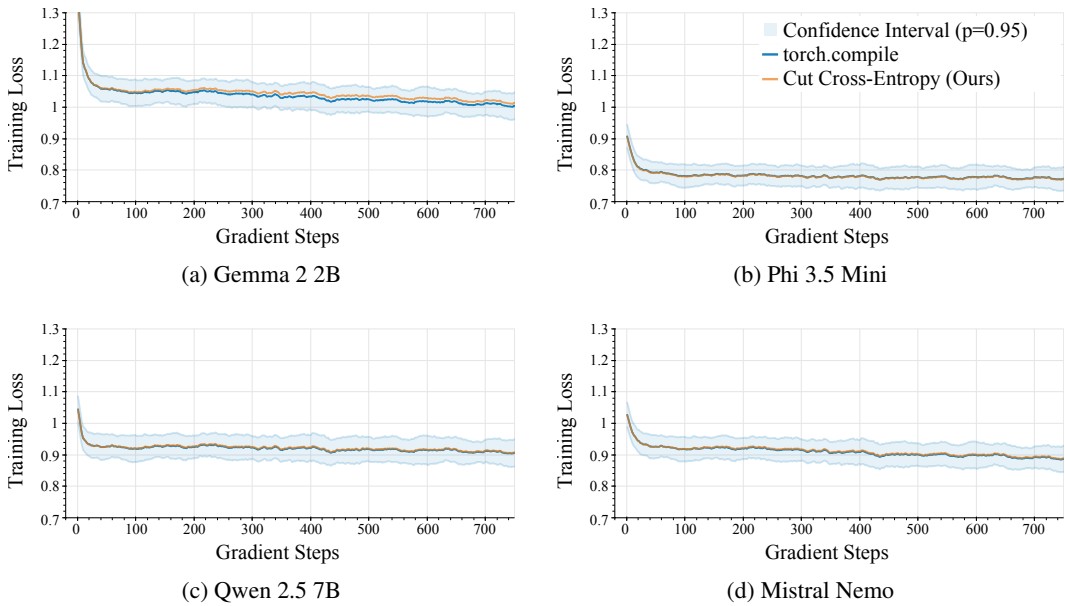

Figure 4: Training loss curves for four models on the Alpaca dataset (Taori et al., 2023). The loss curves for CCE and `torch.compile` are nearly indistinguishable, showing that the gradient filtering in CCE does not impair convergence. Results averaged over 5 seeds.

## 5.2 GRADIENT FILTERING

Fig. 3 shows the sorted softmax probability of vocabulary entries. Note that the probabilities vanish very quickly and, for the top $10^5$ most likely tokens, there is a linear relationship between $\log$ rank and $\log$ probability. Second, by the $\sim$50th most likely token, the probability has fallen bellow our threshold for gradient filtering.

This explains why we are able to filter so many values from the gradient computation without affecting the result. At these sparsity levels, most blocks of the softmax matrix $\mathbf{S}$ are empty.

## 5.3 TRAINING STABILITY

**Fine-tuning.** We fine-tune Qwen 2.5 7B Instruct (Qwen Team, 2024), Phi 3.5 Mini Instruct (Abdin et al., 2024), Gemma 2 2B Instruct (Rivière et al., 2024), and Mistral NeMo (Mistral AI Team, 2024) on the Alpaca Dataset (Taori et al., 2023) using CCE and `torch.compile` as the control. CCE and `torch.compile` have indistinguishable loss curves, demonstrating that the gradient filtering in CCE does not impair convergence (Fig. 4).

**Pretraining.** In our initial experiments using CCE for pretraining, we found that validation perplexity suffered due to two sources of error. First, gradient filtering when applied to $\nabla C$ causes no gradient to be propagated to tokens that have little to no support in the training set. This does not cause issues when fine-tuning but does when pretraining. Second, CCE performs a summation in global memory. It is most efficient to perform this reduction in the desired final floating point type. In pretraining, the resulting loss of precision reduces performance. We use Kahan summation (Kahan, 1965) to recover this loss of precision. This changes correspond to CCE-Kahan-FullC.

We pretrain Qwen 2.5 7B Instruct (Qwen Team, 2024), Phi 3.5 Mini Instruct (Abdin et al., 2024), Gemma 2 2B Instruct (Rivière et al., 2024), and Mistral NeMo (Mistral AI Team, 2024) on the 5% of the Open WebText Dataset (Gokaslan et al., 2019) using CCE-Kahan-FullC and `torch.compile`. We report validation perplexity on a held-out 0.25% of Open WebText and find that CCE-Kahan-FullC produces identical curves as `torch.compile` (Fig. 5).

We make two notes about CCE-Kahan-FullC. First, the increased memory usage of CCE-Kahan-FullC vs. CCE is due to temporary buffers used in the backward pass. The size of these buffers

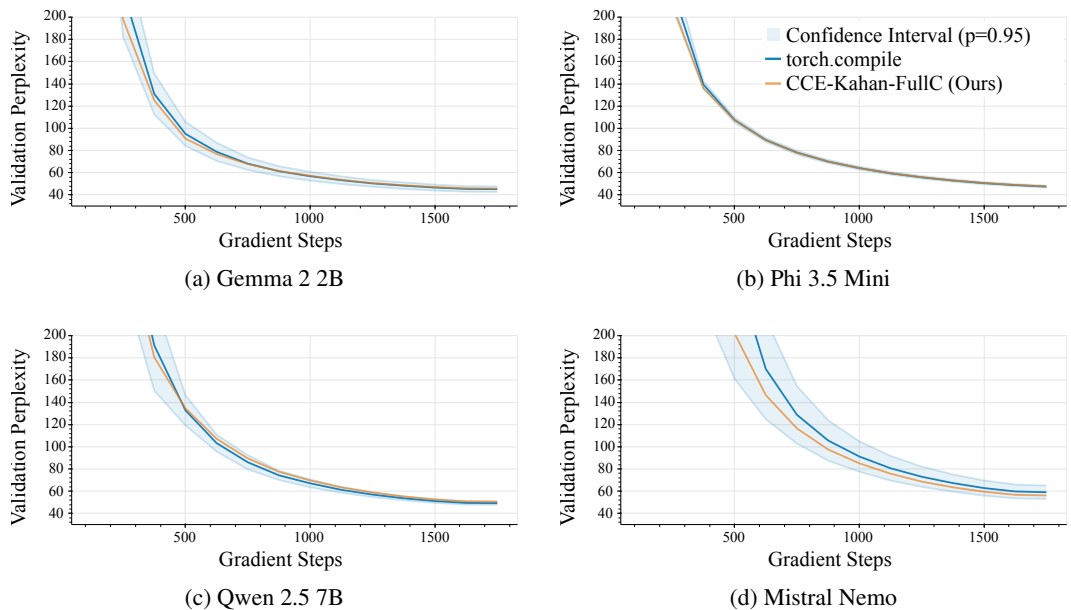

Figure 5: Validation perplexity curves for four models on trained using 5% of the Open WebText dataset (Gokaslan et al., 2019). The validation set is a 0.25% subset of Open WebText that does not overlap with the train set. We find that CCE-Kahan-FullC matches `torch.compile`. Results averaged over 5 seeds.

is typically less than the amount of free memory needed to rematerialize activations when using activation/gradient checkpoint (Chen et al., 2016). Thus CCE-Kahan-FullC often shares the same memory saving benefits as CCE. Second, the increased computation time of CCE-Kahan-FullC vs. `torch.compile` is often offset by the larger batch sizes CCE-Kahan-FullC enables. In our experiments with Mistral NeMo, CCE-Kahan-FullC enabled doubling the batch size, thereby decreasing training time by 2 hours (16%) compared to `torch.compile`.

## 6   DISCUSSION

As vocabulary size $|V|$ has grown in language models, so has the memory footprint of the loss layer. The memory used by this one layer dominates the training-time memory footprint of many recent language models. We described CCE, an algorithm to compute $\ell_i = \log \mathrm{softmax}_i(\mathbf{C}^T f(x_1 \dots x_{i-1}))$ and its gradient with negligible memory footprint.

Beyond the immediate impact on compact large-vocabulary LLMs, as illustrated in Fig. 1, we expect that CCE may prove beneficial for training very large models. Specifically, very large models are trained with techniques such as pipeline parallelism (Huang et al., 2019; Narayanan et al., 2019). Pipeline parallelism works best when all stages are equally balanced in computation load. Achieving this balance is easiest when all blocks in the network have similar memory-to-computation ratios. The classification head is currently an outlier, with a disproportionately high memory-to-computation ratio. CCE may enable better pipeline balancing or reducing the number of stages.

We implemented CCE using Triton (Tillet et al., 2019). Triton creates efficient GPU kernels and enables rapid experimentation but has some limitations in control flow. Specifically, the control flow must be specified at the block level and therefore our thread-safe log-add-exp and gradient filtering are constrained to operate at the block level as well. We expect that implementing CCE in CUDA may bring further performance gains because control flow could be performed at finer-grained levels.

It could also be interesting to extend CCE to other classification problems where the number of classes is large, such as image classification and contrastive learning.

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

## A  Notation

Throughout the paper, we use the following notation conventions. Matrices are bold, capital letters, e.g., $\mathbf{A}$. Indexed matrices are capital letters and are indexed by column and then, optionally, row. For example, given $\mathbf{A} \in \mathbb{R}^{N \times M}$, then e.g., $A_j$ is the length $N$ vector that is the $j$th column for A, $A_{j,i}$ is then the $i$th value in the vector $A_j$. When we combine indexing and transposing, we always index and then transpose.

Vectors are bold lower-case letters, e.g., $\mathbf{x}$, with the exception of LSE which is the vector containing the log-sum-exp (LSE). Indexed vectors are lower-case letters, $x_i$.

In addition to scalar indexing, we also block index matrices when describing how our algorithms are implemented. In these cases, the matrix and vector will maintain their bold to indicate that the indexing refers to a block and thus are still a matrix or vector.

| Notation | Description |
|---|---|
| $\mathbf{E}$ | A $D \times N$ matrix containing batch of inputs. |
| $E_i$ | A $D$-dimensional vector containing the embedding for the $i$th input. |
| $\mathbf{C}$ | A $D \times |V|$ classifier matrix used to compute the logit for each token. |
| $C_i$ | A $D$-dimensional vector used to create the logit for the $i$th token. |
| $\mathbf{x}$ | A length $N$ vector containing the inputs. |
| $x_i$ | A scalar that is the $i$th input. |
| $\mathbf{C}_{x_i}$ | A length $D$ containing the vector used to create the logit for the $x_i$th token. |
| $\mathbf{C}^\top \mathbf{E}$ | A $|V| \times N$ matrix containing the logits over the vocabulary for each input. |
| $\left(\mathbf{C}^\top \mathbf{E}\right)_{\mathbf{x}}$ | A length $N$ vector where the $ith$ entry is the logit for the $x_i$th token. |
| LSE | A length $N$ vector containing the log-sum-exp (LSE) for each input over the vocabulary. |
| $\mathbf{E}_n$ | The $n$th $D \times N_B$ block of $\mathbf{E}$. |
| $\mathbf{E}_{n,d}$ | The $d$th $D_B \times N_B$ block of $\mathbf{E}_n$. |
| $[[\mathbf{a} = \mathbf{b}^\top]]$ | An indicator matrix where the value at the $i$th column and $j$th row is 1 if $a_j = b_i$ and 0 otherwise. |

## B  Removing Ignored Tokens

It is common to have tokens that have no loss computation when training LLMs in practice. Examples include padding, the system prompt, user input, *etc.*. While these tokens must be processed by the backbone – to enable efficient batching in the case of padding or to give the model the correct context for its prediction in the case of system prompts and use inputs – they do not contribute directly to the loss.

In all implementations we are aware of, the logits and loss for these ignored tokens is first computed and then set to zero. We notice that this is unnecessary. These tokens can be removed *before* logits+loss computation with no change to the loss/gradient and save a significant amount of computation.

Table A1 shows the performance of all methods in Table 1 with a filter that removes ignored tokens before logits+loss computation. This represents a significant speed up for all methods but Liger Kernels. Due to heavy chunking in Liger Kernels to save memory, it is bound by kernel launch overhead, not computation, and therefore reducing the amount of computation does not increase speed. Filtering ignored tokens is also a significant memory saving for most all but CCE (because CCE already uses the minimum amount of memory possible).

## C  Additional Results

### C.1  Further Performance Analysis

Table A2 shows a breakdown of the time spent for different components of in the backward pass of CCE and Baseline. For CCE, we selectively disabled/enabled portions of the kernel and measured the time saved to determine the amount of time taken by that component. For Baseline, we manually implemented each operation of the backward pass and timed them seperately.

---

[3]The gradient and loss are computed simultaneously, not in separate forward/backward passes.

| | Method | Loss | | Gradient | | Loss+Gradient | |
|---|---|---|---|---|---|---|---|
| | | Memory | Time | Memory | Time | Memory | Time |
| | Lower bound | 0.004 MB | | 1,161 MB | | 1,161 MB | |
| 1) | CCE (Ours) | **245 MB** | **17 ms** | 1,163 MB | 37 ms | **1,164 MB** | 54 ms |
| 2) | Liger Kernels (Hsu et al., 2024)[3] | 1,316 MB | 301 ms | | | 1,314 MB | 303 ms |
| 3) | Torch Tune Team (2024) (8 chunks) | 3,688 MB | 23 ms | 2,789 MB | 54 ms | 6,157 MB | 77 ms |
| 4) | torch.compile | 1,847 MB | 19 ms | 5,490 MB | **34 ms** | 7,337 MB | **53 ms** |
| 5) | Baseline | 10,997 MB | 30 ms | 7,320 MB | 44 ms | 12,826 MB | 75 ms |
| 6) | CCE (No Vocab Sorting) | 0.06 MB | 17 ms | 1,162 MB | 43 ms | 1,163 MB | 60 ms |
| 7) | CCE (No Grad. Filter) | 0.06 MB | 17 ms | 1,163 MB | 110 ms | 1,163 MB | 126 ms |
| 8) | CCE-Kahan | 1 MB | 18 ms | 2,325 MB | 42 ms | 2,327 MB | 59 ms |
| 9) | CCE-Kahan-FullC | 1 MB | 18 ms | 2,326 MB | 98 ms | 2,327 MB | 114 ms |
| 10) | CCE-Kahan-FullE | 1 MB | 18 ms | 2,325 MB | 92 ms | 2,327 MB | 109 ms |

Table A1: Table 1 where all methods include a filter that removes tokens that are ignored in loss computation. This simple change represents large improvements in practice. Results averaged over 5 seeds.

| Component | Baseline | CCE |
|---|---|---|
| $logits = softcap\left(\mathbf{C}^\top\mathbf{E}\right)$ recomputation | | 45 ms (43.2 %) |
| $\nabla\log softmax_\mathbf{x}\left(logits\right)$ | 35 ms (28.5 %) | 4.7 ms (4.4 %) |
| Gradient Filter | | 1.3 ms (1.2 %) |
| $\nabla softcap\left(\mathbf{C}^\top\mathbf{E}\right)$ | 17 ms (13.7 %) | 4.7 ms (4.4 %) |
| $\nabla\mathbf{E}$ | 37 ms (30.0 %) | 31 ms (29.6 %) |
| $\nabla\mathbf{C}$ | 34 ms (27.7 %) | 18 ms (17.3 %) |

Table A2: Performance breakdown for the backward pass of CCE and Baseline. Gemma 2 (2 B) model. Batch of 8192 tokens. Alpaca dataset used to generate inputs.

CCE spends considerably less time on the cross-entropy loss and softcap portions of the gradient computation. For Baseline, these are very memory intensive operations as there is relatively very little computation done compared the amount of reading/writing. For CCE, the logits are already in SRAM (they were just recomputed) and CCE does not write the result of this computation to main memory, saving a significant amount of time.

Coincidentally, CCE spends a very similar amount of time computing the gradient wrt. the embeddings. CCE spends less time computing the gradient wrt. the classifier. This is because the axis we reduce along for the classifier, N, is shorter than the axis for the embeddings, —V—, and thus leads to less contention on global memory.

Compared to Baseline, CCE saves 30 ms on the gradient of the logits wrt. cross-entropy loss, 12 ms on the gradient wrt. softcapping, 5 ms on the gradient wrt. E, and 15 ms on the gradient wrt. C. This saving of 62 ms more than offsets the 45 ms spent re-computing and applying the gradient filter.

## C.2 ADDITIONAL RUNTIME AND MEMORY

Table A3 shows additional results for Gemma 2 (9 B), Gemma 2 (27 B), Qwen 2.5 (7 B) (Qwen Team, 2024), Qwen 2.5 (32 B), PHI 3.5 Mini (Abdin et al., 2024), and Mistral NeMo (Mistral AI Team, 2024) in the same setting as Table 1. For each model CCE is able to reduce the total memory consumed by the loss by an order of magnitude from the baseline. For forward (Loss) and backward (Gradient) passes combined, CCE is within 3 MB of the lowest possible memory consumption. Compared to Gemma 2 (2 B) all these models have a smaller ratio of the vocabulary size to hidden dimension. This has two impacts.

First, the number of tokens that have a significant gradient is largely constant (it is dependent on the data type). Therefore proportionally less of the gradient will be filtered out.

Second, for all other methods increasing the hidden dimension increase the amount of parallelism that can be achieved. Liger Kernels (Hsu et al., 2024) sets its chunk size based on $|V|/D$ – the lower that ratio, the bigger the chunk size. As $|V|/D$ continues to decrease, Liger Kernels is able to make better use of the GPU. All other methods use two matrix multiplications to compute the gradient. The amount of work that can be performed in parallel to compute $\nabla E$ and $\nabla C$ is $B \times D$ and $|V| \times D$, respectively[4]. The amount of parallel work for CCE is $B \times |V|$, thus increasing $D$ increases the amount of work but not the amount of parallelism. It may be possible leverage ideas from split-k matrix multiplication kernels to expose more parallelism to CCE for large values of $D$.

For the smallest $|V|/D$ considered, Phi 3.5 Mini ($|V|$=32,064, D=3,072) ours is approximately 50% slower (12 ms) than `torch.compile` (although it uses substantially less memory). In our experiments, this increase in linear-cross-entropy loss computation time is largely negligible and only increases training time by one to two percent.

We also consider how changing the number of tokens changes performance (Figs. A1 and A2). We find that CCE behaves very similarly to Baseline and `torch.compile`. Further, because CCE does not utilize chunking, it does not reach a point where the overhead of dispatching all the kernels becomes the dominating factor. We also find that while CCE-Kahan-FullC is slower than the Liger Kernel and Torch Tune baselines with a large number of tokens, it becomes more performant than those baselines as the number of tokens reduces.

## D    MEMORY USE METHOD DETAILS

Table A4 contains the raw numbers used to create Fig. 1. The maximum batch size for 16 GPUs was calculated by assuming that the total amount of memory available is $75 \times 16$ (i.e., each 80 GB GPU will be fully occupied expect for a 5 GB buffer for various libraries), then subtracting the memory used for weights + optimizer + gradients and then diving by the memory used per token.

The numbers in Table A4 are computed using the following methods. When present, the number of tokens is assumed to be 65,536.

We compute the amount of memory used for intermediate activations as the number of layers times the hidden size times number of tokens times 2 bytes per bfloat16. This assumes the use of activation/gradient checkpointing (Chen et al., 2016) for transformer layer.

The amount of memory used by the logits is the number of tokens times the vocabulary size times 4 bytes per float32. This likely undercounts the amount of memory used for computing the probability distribution, as its common to also keep a copy of the logits in bfloat16 and, for models like Gemma 2 (Rivière et al., 2024) that use logit softcapping, an additional copy of the logits after softcapping may be needed. However, this method can be uniformly applied to all models.

The amount of memory used by Weights+Opt+Grad is the number of parameters times 4 (parameters, gradient, and Adam first and second moments) times 2 bytes per bfloat16.

## E    FLOATING POINT ADDITION

Here we provide a brief explanation of floating point addition and how it relates to our proposed gradient filtering.

Given two numbers $a$ and $b$ represented using floating point, such that $|a| < |b|$, the following steps are performed

1. Separate the mantissa (the fractional part) and the exponent from both numbers $a$ and $b$.
2. Re-write the mantissa of the smaller number ($a$ in our case) such that it shares the same exponent as the $b$.
3. Add the re-written mantissa of $a$ to the mantissa of $b$.

---

[4]Ignoring split-k matrix multiplication kernels for simplicity.

| Method | Loss | | Gradient | | Loss+Gradient | |
|---|---|---|---|---|---|---|
| | Memory | Time | Memory | Time | Memory | Time |
| **Gemma 2 (9 B)** (Rivière et al., 2024) ($|V|$=256,000, D=3,584) | | | | | | |
| Lower bound | 0.004 MB | | 1,806 MB | | 1,806 MB | |
| CCE (Ours) | **1 MB** | **68 ms** | **1,808 MB** | 141 ms | **1,809 MB** | 208 ms |
| Liger Kernels (Hsu et al., 2024) | 2,119 MB | 418 ms | | | 2,119 MB | 419 ms |
| Torch Tune Team (2024) (8 chunks) | 8,000 MB | 75 ms | 3,264 MB | 168 ms | 11,264 MB | 243 ms |
| torch.compile | 4,000 MB | 70 ms | 12,000 MB | **134 ms** | 16,000 MB | **207 ms** |
| Baseline | 24,000 MB | 102 ms | 16,000 MB | 164 ms | 28,000 MB | 271 ms |
| CCE-Kahan-FullC | 1 MB | 68 ms | 3,558 MB | 384 ms | 3,559 MB | 450 ms |
| **Gemma 2 (27 B)** (Rivière et al., 2024) ($|V|$=256,000, D=4,608) | | | | | | |
| Lower bound | 0.004 MB | | 2,322 MB | | 2,322 MB | |
| CCE (Ours) | **1 MB** | **83 ms** | **2,324 MB** | 200 ms | **2,325 MB** | 281 ms |
| Liger Kernels (Hsu et al., 2024) | 2,948 MB | 361 ms | | | 2,948 MB | 363 ms |
| Torch Tune Team (2024) (8 chunks) | 8,000 MB | 91 ms | 4,768 MB | 204 ms | 12,768 MB | 296 ms |
| torch.compile | 4,000 MB | 86 ms | 12,000 MB | **168 ms** | 16,000 MB | **256 ms** |
| Baseline | 24,000 MB | 119 ms | 16,000 MB | 197 ms | 28,000 MB | 322 ms |
| CCE-Kahan-FullC | 1 MB | 83 ms | 4,574 MB | 513 ms | 4,575 MB | 593 ms |
| **Mistral NeMo** (Mistral AI Team, 2024) ($|V|$=131,072, D=5,120) | | | | | | |
| Lower bound | 0.004 MB | | 1,360 MB | | 1,360 MB | |
| CCE (Ours) | **0.6 MB** | 52 ms | **1,361 MB** | 129 ms | **1,362 MB** | 180 ms |
| Liger Kernels (Hsu et al., 2024) | 1,872 MB | 166 ms | | | 1,872 MB | 167 ms |
| Torch Tune Team (2024) (8 chunks) | 2,048 MB | 49 ms | 3,348 MB | 113 ms | 5,396 MB | 161 ms |
| torch.compile | 2,048 MB | **48 ms** | 6,144 MB | **94 ms** | 8,192 MB | **143 ms** |
| Baseline | 10,240 MB | 58 ms | 8,192 MB | 100 ms | 12,288 MB | 161 ms |
| CCE-Kahan-FullC | 0.6 MB | 52 ms | 2,641 MB | 291 ms | 2,642 MB | 342 ms |
| **Phi 3.5 Mini** (Abdin et al., 2024) ($|V|$=32,064, D=3,072) | | | | | | |
| Lower bound | 0.004 MB | | 236 MB | | 236 MB | |
| CCE (Ours) | **0.2 MB** | 8 ms | **236 MB** | 26 ms | **236 MB** | 34 ms |
| Liger Kernels (Hsu et al., 2024) | 487 MB | 26 ms | | | 488 MB | 26 ms |
| Torch Tune Team (2024) (8 chunks) | 502 MB | 9 ms | 451 MB | 18 ms | 953 MB | 30 ms |
| torch.compile | 502 MB | **8 ms** | 1,504 MB | **15 ms** | 2,006 MB | **22 ms** |
| Baseline | 2,506 MB | 11 ms | 2,004 MB | 16 ms | 3,006 MB | 27 ms |
| CCE-Kahan-FullC | 0.2 MB | 8 ms | 424 MB | 46 ms | 424 MB | 54 ms |
| **Qwen 2.5 (7 B)** (Qwen Team, 2024) ($|V|$=152,064, D=3,584) | | | | | | |
| Lower bound | 0.004 MB | | 1,096 MB | | 1,096 MB | |
| CCE (Ours) | **0.6 MB** | 43 ms | **1,098 MB** | 93 ms | **1,097 MB** | 136 ms |
| Liger Kernels (Hsu et al., 2024) | 1,394 MB | 171 ms | | | 1,394 MB | 171 ms |
| Torch Tune Team (2024) (8 chunks) | 2,379 MB | 42 ms | 2,540 MB | 96 ms | 4,921 MB | 138 ms |
| torch.compile | 2,376 MB | **41 ms** | 7,128 MB | **79 ms** | 9,504 MB | **121 ms** |
| Baseline | 11,880 MB | 53 ms | 9,504 MB | 86 ms | 14,256 MB | 142 ms |
| CCE-Kahan-FullC | 0.6 MB | 43 ms | 2,138 MB | 225 ms | 2,138 MB | 267 ms |
| **Qwen 2.5 (32 B)** (Qwen Team, 2024) ($|V|$=152,064, D=5,120) | | | | | | |
| Lower bound | 0.004 MB | | 1,565 MB | | 1,565 MB | |
| CCE (Ours) | **0.6 MB** | 60 ms | **1,566 MB** | 133 ms | **1,567 MB** | 193 ms |
| Liger Kernels (Hsu et al., 2024) | 2,159 MB | 192 ms | | | 2,161 MB | 192 ms |
| Torch Tune Team (2024) (8 chunks) | 2,376 MB | 57 ms | 3,882 MB | 130 ms | 6,259 MB | 186 ms |
| torch.compile | 2,376 MB | **56 ms** | 7,128 MB | **108 ms** | 9,504 MB | **165 ms** |
| Baseline | 11,880 MB | 68 ms | 9,504 MB | 115 ms | 14,256 MB | 186 ms |
| CCE-Kahan-FullC | 0.6 MB | 61 ms | 3,052 MB | 326 ms | 3,053 MB | 384 ms |

Table A3: Memory usage and time of CCE, Liger Kernels, Torch Tune, torch.compile, and Baseline for additional models. Batch of $8,192$ tokens. Results averaged over 5 seeds.

> 4. Combine the resulting mantissa and exponent of $b$ and then convert them into normalized form.

Step 2 is where truncation happens and the intuition of gradient filtering comes from. In bfloat16, if the exponent of $b$ is more than $2^7$ times larger than that of a, the 7-bit mantissa no longer has enough precision to represent any of $a$'s mantissa and in the process of re-writing, $a$ will be, in effect, set to zero. For gradient filtering, we are only concerned with values in the range $[0, 1]$, so the threshold of $2^{-12}$ means that we only keep values that don't get rounded to zero when $b = 2^{-5}$.

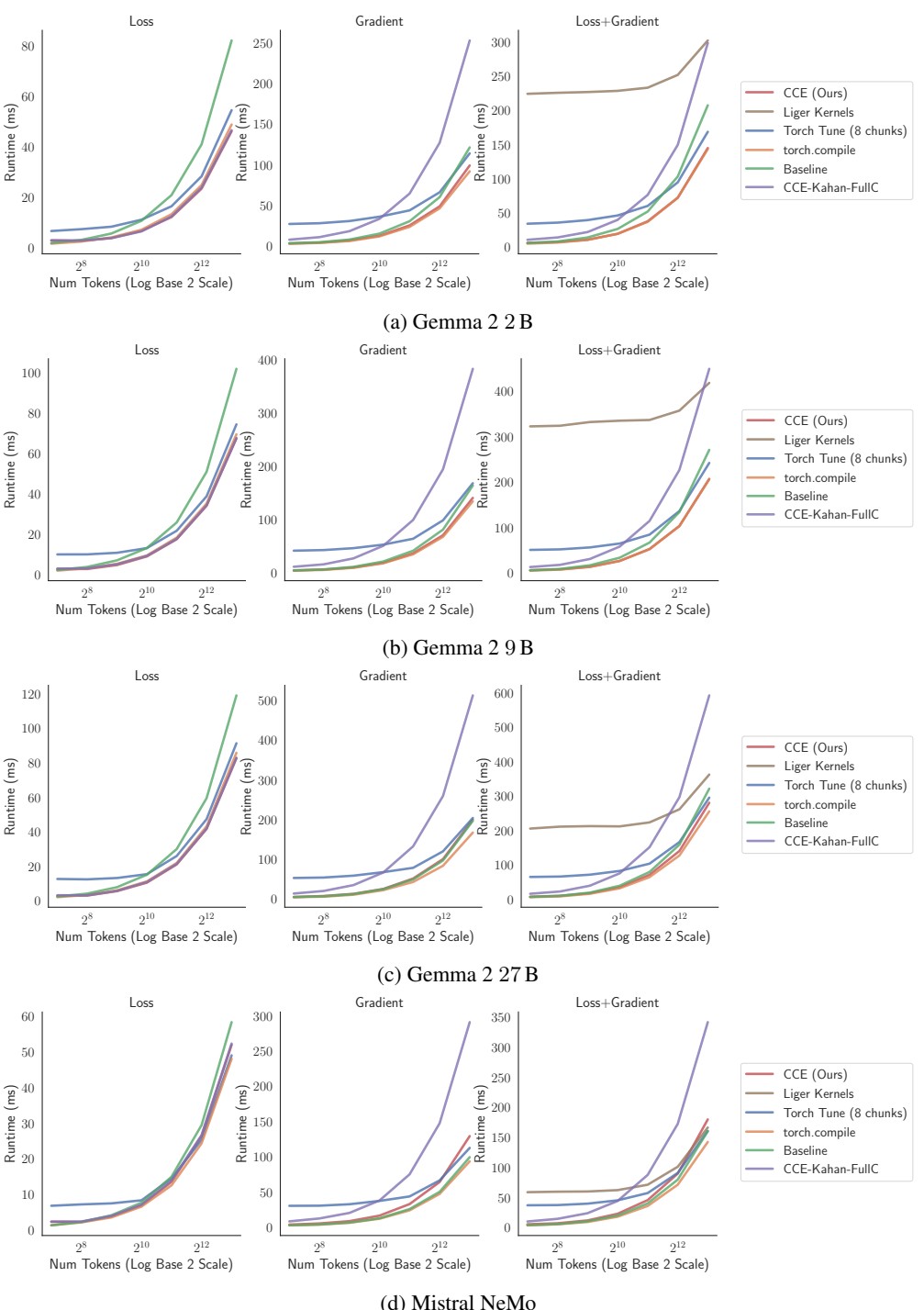

Figure A1: Performance of CCE and baselines for all models with a varying batch sizes. Results averaged over 5 seeds. Continued in Fig. A2.

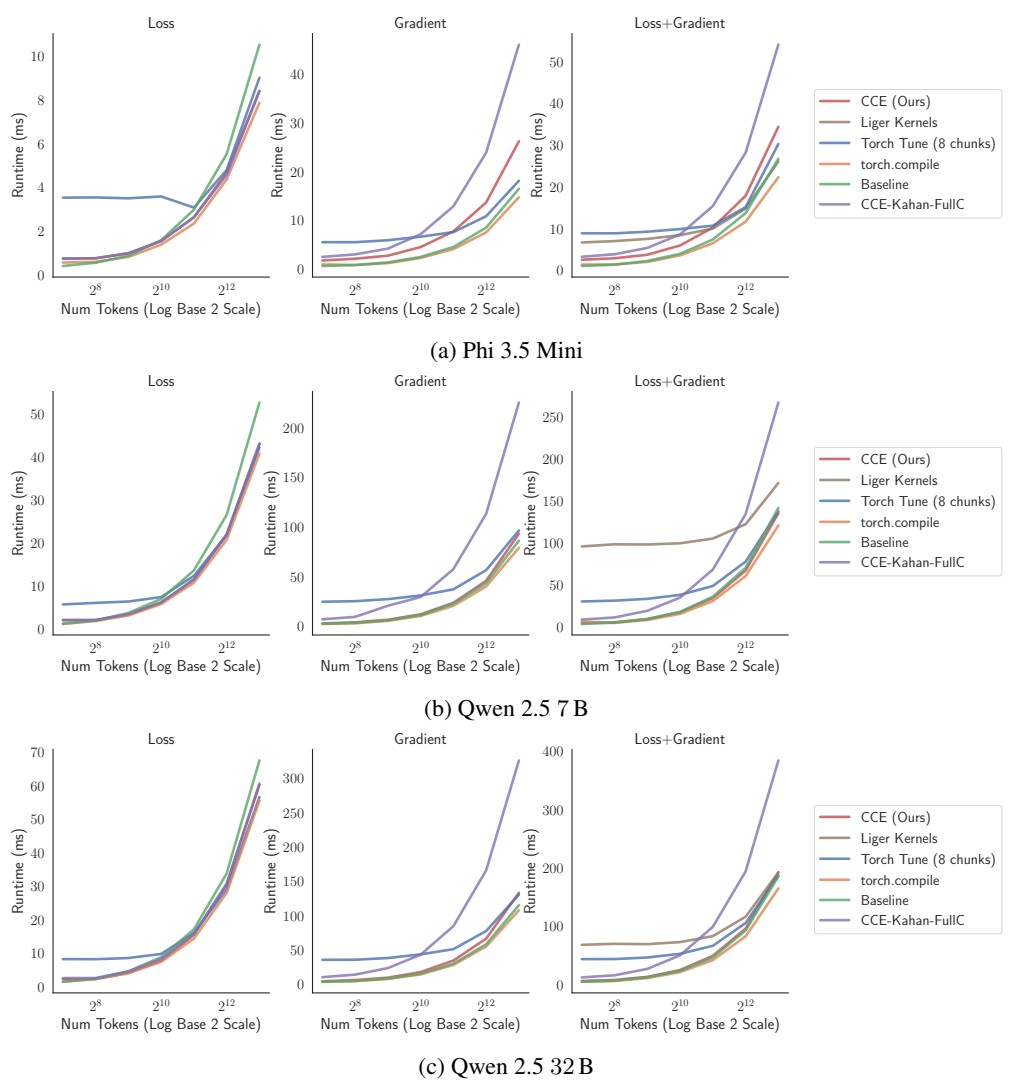

Figure A2: Performance of CCE and baselines for all models with a varying batch sizes. Results averaged over 5 seeds.

| Model | Logits | Activations | Weights+Opt+Grad | Max Batch Size (Before) | Max Batch Size (After) | Increase |
|---|---|---|---|---|---|---|
| GPT 2 | 12,564 MB | 1,152 MB | 1,045 MB | 5,866,190 | 69,845,595 | 11.9× |
| GPT Neo (1.3 B) | 12,564 MB | 6,144 MB | 10,421 MB | 4,268,047 | 12,996,042 | 3.0× |
| GPT Neo (2.7 B) | 12,564 MB | 10,240 MB | 20,740 MB | 3,471,784 | 7,731,585 | 2.2× |
| Gemma (2 B) | 64,000 MB | 4,608 MB | 19,121 MB | 1,155,515 | 17,204,330 | 14.9× |
| Gemma 2 (27 B) | 64,000 MB | 26,496 MB | 207,727 MB | 739,448 | 2,525,554 | 3.4× |
| Gemma 2 (2 B) | 64,000 MB | 7,488 MB | 19,946 MB | 1,108,206 | 10,580,057 | 9.5× |
| Llama 2 (13 B) | 8,000 MB | 25,600 MB | 99,303 MB | 2,203,057 | 2,891,512 | 1.3× |
| Llama 2 (7 B) | 8,000 MB | 16,384 MB | 51,410 MB | 3,164,429 | 4,709,560 | 1.5× |
| Llama 3 (70 B) | 32,064 MB | 81,920 MB | 538,282 MB | 397,019 | 552,414 | 1.4× |
| Llama 3 (8 B) | 32,064 MB | 16,384 MB | 61,266 MB | 1,579,333 | 4,670,136 | 3.0× |
| Mistral 7 B | 8,000 MB | 16,384 MB | 55,250 MB | 3,154,108 | 4,694,200 | 1.5× |
| Mixtral 8x7 B | 8,000 MB | 16,384 MB | 356,314 MB | 2,344,949 | 3,489,944 | 1.5× |
| Phi 1.5 | 12,574 MB | 6,144 MB | 10,821 MB | 4,264,482 | 12,991,781 | 3.0× |
| Phi 3 Medium | 8,003 MB | 25,600 MB | 106,508 MB | 2,188,824 | 2,873,067 | 1.3× |
| Qwen 1.5 (7 B) | 37,912 MB | 16,384 MB | 58,909 MB | 1,412,087 | 4,679,564 | 3.3× |

Table A4: Raw data for Fig. 1. Memory usage calculated using a global batch size of 65,536.

---

**Algorithm 4** Memory-efficient linear-cross-entropy loss, backward pass

---

**Inputs:**    $\mathbf{E} \in \mathbb{R}^{D \times N}$, $\mathbf{C} \in \mathbb{R}^{D \times |V|}$, $\mathrm{LSE} \in \mathbb{R}^N$, $\nabla \mathrm{CEL} \in \mathbb{R}^N$, and $\mathbf{x} \in \mathbb{R}^N$.
                    Block sizes $N_B$, $V_B$, and $D_B$.
                    Accuracy threshold $\varepsilon$.
                    $\mathbf{v} = [1, \ldots, |V|]$.
**Outputs:**   $\nabla \mathbf{E} \in \mathbb{R}^{D \times N}$, $\nabla \mathbf{C} \in \mathbb{R}^{D \times |V|}$

---

**for** all pairs of blocks $\mathbf{E}_n$, $\mathbf{C}_v$ **do**    ▷ Divide $\mathbf{E}$ and $\mathbf{C}$ into blocks of size $D \times N_B$ and $D \times V_B$
    $\mathbf{A}_{nv} = \mathbf{0}_{V_B \times N_B}$    ▷ Zero matrix of size $V_B \times N_B$ in on-chip SRAM
    **for** blocks $\mathbf{E}_{n,d}$, $\mathbf{C}_{v,d}$ **do**    ▷ Divide $\mathbf{E}_n$ and $\mathbf{C}_v$ into blocks of $D_B \times N_B$ and $D_B \times V_B$
        $\mathbf{A}_{nv} \mathrel{+}= \mathbf{C}_{v,d}^{\top} \cdot \mathbf{E}_{n,d}$    ▷ Blockwise matrix multiplication
    **end for**
    $\mathbf{S}_{nv} = \exp(\mathbf{A}_{nv} - \mathrm{LSE}_n)$    ▷ Compute the softmax
    $\mathbf{G}_{nv} = \left[\!\left[\mathbf{v}_v = \mathbf{x}_n^{\top}\right]\!\right] - \mathbf{S}_{nv}$    ▷ Gradient of cross-entropy loss wrt. logits
    **if** all$(|\mathbf{G}_{nv}| < \varepsilon)$ **then**
        **skip**    ▷ Skip computation if below desired numerical precision
    **end if**
    **for** blocks $\mathbf{E}_{n,d}$, $\mathbf{C}_{v,d}$ **do**    ▷ Divide $\mathbf{E}_n$ and $\mathbf{C}_m$ into blocks of $D_B \times N_B$ and $D_B \times V_B$
        $\nabla \mathbf{E}_{n,d}^{\top} \mathrel{+}= (\mathbf{G}_{nv} \cdot \nabla \mathrm{CEL}_n)\, \mathbf{C}_{v,d}$    ▷ Locking thread-safe gradient update
        $\nabla \mathbf{C}_{v,d}^{\top} \mathrel{+}= (\mathbf{G}_{nv} \cdot \nabla \mathrm{CEL}_n)^{\top} \mathbf{E}_{n,d}$    ▷ Locking thread-safe gradient update
    **end for**
**end for**

