# OpenReview forum: "Cut Your Losses in Large-Vocabulary Language Models"
_ICLR.cc/2025/Conference — ICLR 2025 Oral_

### Official Review · Reviewer_HR95 · 2024-10-31

**Soundness:** 3
**Presentation:** 3
**Contribution:** 4
**Rating:** 10
**Confidence:** 3

**Summary:**

This work proposes Cut Cross-Entropy (CCE) to address the massive memory footprint of the standard cross-entropy loss calculation in LLM training. CCE tiles and fuses the logits indexing and the matmul between tiles of `lm_head` and embeddings in the forward pass. In backward propagation, CCE introduces gradient filtering and vocabulary sorting to optimize memory access pattern with negligible approximation errors. CCE presents experiments showing the effectively reduced memory footprint and indistinguishable influence on fine-tuning.

**Strengths:**

- This paper identifies a new challenge brought by the large size of vocabulary of language models, especially LLMs, i.e., the massive memory footprint consumed by the cross-entropy loss computation.
- The tricks of gradient filtering and vocabulary sorting in the proposed method are enlightening.
- The author implemented CUDA kernels to support the algorithm and provides experiments to verify the reduced memory footprint and latency.

**Weaknesses:**

- The symbols in the derivation of CCE can be clearer. For example, the symbols in page 4 between line 186 and 215, such as $C^T$, $C_{x_i}^T$, $C_X$, and $(C^T E)_X$, look confusing at first glance. It may be helpful to have a table of symbol definition in the appendix.
- Experiments on how the memory and latency of CCE kernel varies with the vocab size & model family can be added.
  - Current Tab 1 presents the memory and latency results of Gemma-2-2B.  The vocab size of Gemma-2-2B is 256000, which is larger than other LLMs. For example, the vocab size is 128256 for Llama-3-8B/70B/405B, 32768 for mistral-7B-v0.3, and 32064 for Phi-3.5. If the size of `lm_head` is `(model_hidden_size, vocab_size)`. When the `model_hidden_size` increases, do we expect a diminishing benefit of CCE? The evaluation will be more comprehensive if the author could discuss:
  - Compared to the baselines, how the memory and latency change if CCE is applied to Gemma-2-27B training (same vocab size as Tab 1, but larger model hidden size)
  - Compared to the baselines, how the memory and latency change if CCE is applied to training of the models like Phi3.5-mini (smaller vocab size, similar model size).

**Questions:**

- Current Fig. 4 verifies that CCE has negligible influence on LLM fine-tuning. I am also curious about the impact of CCE on LLM pretraining. If an LLM is trained from scratch using CCE, how will the randomly initialized weights influence the gradient filtering and vocab sorting?

For other questions, please refer to my concerns in the Weakness section.

---

> ### Author Response · Authors · 2024-11-25
> **Respond to HR95**
>
> We thank the reviewer for their review. We are pleased they found that our work “identifies a new challenge brought by the large size of vocabulary of language models”, that are proposed gradient filtering and vocabulary sorting are “enlightening”, and our experiments show the “reduced memory footprint and indistinguishable influence on fine-tuning.”
>
> > The symbols in the derivation of CCE can be clearer. For example, the symbols in page 4 between line 186 and 215, such as $C^T$, $C_{x_i}^T$, $C_X$, and $(C^T E)_X$, look confusing at first glance. It may be helpful to have a table of symbol definition in the appendix.
>
> Thank you for the suggestion! We have added a new Section A the appendix. Let us know if you have any additional suggestions.
>
> > Experiments on how the memory and latency of CCE kernel varies with the vocab size & model family can be added.
>
> We wanted to know this too! We have added benchmarking of Gemma 2 (9 B), Gemma 2 (27 B), Llama 3 (8 B), Mistral NeMo (12 B), and Phi 3.5 mini to the appendix.
>
> We find that as the ration of vocabulary size to hidden size decrease, the latency of CCE increases relative to torch.compile, but it continues to save a large amount of memory. Only for Phi 3.5 mini is CCE slower than torch.compile (11 ms, 50% slower), but it continues to save substantial memory.
>
> While this difference may seem large, in practice it is largely negligible. In our fine-tuning experiments, CCE increases total training time by 1-2% for Phi 3.5 Mini.
>
> > I am also curious about the impact of CCE on LLM pre training.
>
> We have updated the appendix (Section C.1) to include small-scale experiments pre-training Gemma 2 (2B), Llama 3 (8 B), Mistral NeMo (12 B), and Phi 3.5 mini.
>
> CCE has identical training-loss curves to torch.compile. However, CCE results in lower probabilities on tokens that are in the validation set but not in the training set and this results in higher validation perplexities. If we examine validation sequences with only tokens that were seen at training time (but still a novel combination of tokens), then validation perplexity matches torch.compile.
>
> If this effect still persists in full-scale pre-training remains an open question.
>
> > If an LLM is trained from scratch using CCE, how will the randomly initialized weights influence the gradient filtering and vocab sorting?
>
> In our small-scale pre-training experiments, the randomly initialized weights reduced the effectiveness of gradient filtering and vocabulary sorting. The impact of this depends on the model. For Phi 3.5 Mini, CCE increased total training-time by 25% compared to torch.compile, while Gemma 2 saw an increase of less than 1%.
>
> Let us know if we have addressed your concerns.
>
> Finally, if you believe this paper should be highlighted at the conference, could you please consider raising your score to reflect that?

---

> > ### Comment · Reviewer_HR95 · 2024-11-25
> >
> > The additional experiments and explanations have addressed my concern.
> >
> > I would like to raise my score and believe this paper should be highlighted at the conference.

---

### Official Review · Reviewer_dN8N · 2024-11-01

**Soundness:** 3
**Presentation:** 3
**Contribution:** 3
**Rating:** 8
**Confidence:** 4

**Summary:**

This paper proposes Cut Cross-Entropy(CCE) to reduce the memory consumption of Classifier Head and CrossEntropy Loss. They find that the vocabulary of LLMs continuously to grows, and under the gradient checkpointing setting, these part takes more than 50% of the memory consumption. CCE reduce the memory overhead by fusing the classifier head and the calculation of cross entropy loss into 1 kernel, and not materializing the intermediate logits in the forward process. In the backward pass they re-compute the intermediate values to avoid this additional memory overhead (which is quite similar to FlashAttention's design). They further propose to leverage the sparsity pattern in the gradient of classifier head to reduce the amount of computation. CCE reduce the memory overhead by 20x for "Loss+Gradient" part and their loss curve matches the BF16 training baseline.

**Strengths:**

1. The problem this paper trying to solve is well-motivated.
2. The solution to avoid the materialization of the large logit tensor is clear and easy to understand
3. The CCE component is easy to deploy in realistic setting.
4. The performance do not degrade (since the algorithm is nearly lossless considering the high sparsity level)
5. The paper writing is very clear and easy to follow (represent C, E, and LSE in different colors)

**Weaknesses:**

1. "The probabilities materialized by the cross-entropy layer account for 89% of the memory consumption of Gemma 2 for single sequence x with length N = 80000". Can you provide details about how the 89% number is calculated and include a brief calculation or breakdown of the memory usage in the paper or appendix?
2. Does this assumption still hold true when gradient checkpointing = False? I think most of the analysis in this paper is based on the assumption that gradient checkpointing = True. Include a subsection to discuss or analysis of how your method performs when gradient checkpointing is disabled is appreciated.
3. Similar to 2, In Table 1, can you explain where 1477MB, 8000MB, 4000MB, and 24000MB come from? If I understand correctly, the logits.shape is (8192, 256000) in float32, which should take 8000MB memory in total.
4. In Section 4.3, the Gradient filtering paragraph, "If stored in bfloat16 with a 7-bit fraction, any value below 2^{-12} will likely be ignored due to truncation in the summation or rounding in the normalization." Can you explain this in detail? Providing a brief explanation of the numerical precision issues in bfloat16 and how they relate to the gradient filtering threshold is appreciated.

Others:
What LSE stands for (Log-Sum-Exp) should be defined when it is on its first use.

**Questions:**

Please see weakness.

---

> ### Author Response · Authors · 2024-11-25
> **Response to dN8N (1/2)**
>
> We thank the reviewer for their review. We are please they found our problem “well-motivated”, our solution “clear and easy to understand”, and our writing “very clear and easy to follow.”
>
> > Can you provide details about how the 89% number is calculated and include a brief calculation or breakdown of the memory usage in the paper or appendix?
>
> We provide the raw numbers we use in this calculation in the appendix, Table A2 (in the initial version, Table A4 in the updated version). 89% for Gemma2 comes from logits / (logits + activations) = 64000 MB / (64000 MB + 7488 MB) = 89.5%. This ignores the memory used by Weights+Opt+Grad as the amount of memory used by that is unaffected by the number of tokens and depends on the exact details of the sharding strategy and number of GPUs.
>
> We have updated the appendix to provide more detail. To summarize here, we use a simplified model and compute the memory usage as follows:
>
> Logits: NumTokens * VocabularySize * BytesPerFP32
> Activations: NumTokens * HiddenSize * NumLayers * BytesPerBF16
>
> This assumes activation checkpointing after every transformer layer and bfloat16. We assume a global batch size of 65536 tokens (a realistic number of 16 GPUs).
>
> > Memory usage without activation/gradient checkpointing
>
> Without activation/gradient checkpoint, the memory used by intermediate activations would dominate all other sources of memory usage, for example 80+% for Llama 3 (8B). In this case CCE would reduce memory use by 10%. However, training without activation checkpointing is practically infeasible.
>
> > I think most of the analysis in this paper is based on the assumption that gradient checkpointing = True
>
> We depend on activation/gradient checkpointing only when contextualizing the memory consumption of logits relative to the other parts of model training. Other analysis, e.g. Table 1, does not depend on activation/gradient checkpointing.
>
> > How does CCE perform without gradient checkpointing?
>
> CCE is complimentary to activation/gradient checkpoint and does not depend on it. Without activation/gradient checkpointing, CCE would still continue to perform the same and save the same absolute amount of memory, although amount relative to the total memory footprint would decrease substantially due to intermediate activations dominating.
>
> > can you explain where 1477MB, 8000MB, 4000MB, and 24000MB come from? If I understand correctly, the logits.shape is (8192, 256000) in float32, which should take 8000MB memory in total.
>
> Certainly. These numbers come from profiling methods to compute linear-cross-entropy loss in PyTorch and, unlike the model memory footprint numbers, are not calculated. We profiled to show real-world memory usage that accounts for all the realities of buffer re-use, allocator requirements, temporary buffers that are specific to the requirements of that implementation, etc.
>
> 24,000 MB comes from using PyTorch to compute linear-cross-entropy loss. The memory ends up being much higher than just the 8,000 MB to store the logits in float32. In addition to the fp32 logits, we also need the logits in bfloat16 (the result of $C^\top E$, which is performed in bf16), the logits in bf16 after the softcap is applied (Gemma2 specific), and the log-probabilities in float32. These 4 buffers alone account for 24,000 MB.
>
> 4,000 MB comes from using torch.compile to optimize computation. Exactly how torch.compile is able to save this memory is quite opaque. We suspect that it saves memory by fusing kernels, aggressive buffer re-use, and reducing the number of temporary buffers.
>
> 8,000MB comes from using Torch Tune (Torch Tune Team, 2024) with 8 chunks. This performs the computation in chunks and therefore reduces the peak memory utilization as its able to re-use memory.
>
> 1,477MB comes from using Liger Kernels (Hsu et al., 2024). This method makes even heavier use of chunking and adds in custom CUDA kernels to reduce the number of intermediary buffers.

---

> > ### Author Response · Authors · 2024-11-25
> > **Response to dN8N (2/2)**
> >
> > > In Section 4.3, the Gradient filtering paragraph, "If stored in bfloat16 with a 7-bit fraction, any value below 2^{-12} will likely be ignored due to truncation in the summation or rounding in the normalization." Can you explain this in detail? Providing a brief explanation of the numerical precision issues in bfloat16 and how they relate to the gradient filtering threshold is appreciated.
> >
> > Adding two numbers in floating point follows the following logic:
> >
> > Let’s assume we are adding two numbers, a and b, such that b is bigger than a, then
> >
> > Step 1. Separate the mantissa (the fractional part) and the exponent
> > Step 2. Re-write the mantissa of the smaller number (a in our case) such that it shares the same exponent as the larger number
> > Step 3. Add the mantissas of a and b
> > Step 4. Convert the resulting mantissa and exponent into normalized form.
> >
> > Step 2 is where truncation happens and the intuition of gradient filtering comes from. In bfloat16, if the exponent of b is more than 2^7 times larger than that of a, the 7-bit mantissa no longer has enough precision to represent any of a using the exponent of b. For gradient filtering, we are only concerned with values in the range [0, 1], so the threshold of 2^{-12} means that we only keep values that don’t get rounded to zero when b = 2^{-5}.
> >
> > We have added this to the Appendix.
> >
> > Let us know if we have addressed your concerns.
> >
> > Finally, if you believe we have, could you please consider raising your score to reflect that?

---

> > > ### Comment · Reviewer_dN8N · 2024-11-25
> > > **Increase the score to 8**
> > >
> > > Thank the authors for their explanation, and I raise my score accordingly.

---

### Official Review · Reviewer_2Zay · 2024-11-02

**Soundness:** 3
**Presentation:** 2
**Contribution:** 3
**Rating:** 6
**Confidence:** 2

**Summary:**

The paper proposes a novel method of skipping most of the unneeded computation inside LM heads during training when using cross-entropy loss. It's key contributions are:
- A memory efficient indexed matrix multiplication method, which employs sparsity to accelerate the computation.
- A memory efficient linear-log-sum-exp method, which employs dynamic chunking to reduce memory requirements.
- Gradient filtering, which further improves sparsity of the gradient computation.
- Vocabulary sorting, which allows entire chunks to be skipped during computation.

The method is evaulated on both speed, memory usage and convergence, where it shows massive improvements in memory usage for computing losses compared to alternative methods, marginal improvements in speed and negligible degradation in convergence and training quality.

**Strengths:**

- Well motivated problem. Reducing the memory footprint of LLMs during training is important.
- Method generalizes beyond transformer LLMs.
- Demonstrates convergence guarantees compared to cross entropy.
- Extensive benchmark results.

**Weaknesses:**

- Preliminaries (section 3) does not adequately prepare the reader for the complexity of the notation in section 4.

- Section 4 is particularly hard to understand if the reader does not have a deep understanding of GPU kernels and the architecture of modern LLMs.
  - There is a lack of key insights, CCE seems like an arbitrary monolithic algorithm that came out of nowhere.

  - Prehaps decoupling the theoretical reasoning from the actual GPU implementation could make the explanation clearer. For example, in line 201, it says "section 4.2 describes how to compute the [...] operation efficiently", but it is initially unclear to the reader why that operation might be efficient unless the reader can fully understand the intricacies of creating an efficient GPU kernel as described in section 4.2. Same goes for section 4.1 and 4.3.

  - Otherwise, starting from an already efficient GPU implementation of standard CE and focusing on the steps needed to modify it into the CCE method could further improve readability and clarity.

- A lack of ablation studies for the extensive modifications brought on by CCE
  - Section 4.1, 4.2 and 4.3 makes a large number of significant assumptions, modifications and improvements to the traditional CE algorithm, it is not clear whether each modification is actually necessary or which are the most important ones.
  - Unclear whether CCE's improvements is GPU dependent or not. Would it work in non-parallel cases such as on a single-threaded CPU?

**Questions:**

- What are the theoretical justifications on why CCE might be much more computationally and memory efficient compared to traditional CE? For example, why can't you apply the same chunking strategies used in CCE for traditional CE?
- What are the key insights that make CCE work? It seems to me currently that all of the contributions are mixed together, where CCE is an all or nothing monolithic algorithm.
- How does CCE compare to CE on the CPU, or non-parallel cases? Are the improvements algorithmic or does it need to take advantage of GPU parallelization strategies?

---

> ### Author Response · Authors · 2024-11-25
> **Reviewer 2Zay**
>
> We thank the reviewer for their review. We are pleased that the found our problem “well motivated”, that “reducing the memory footprint of LLMs during training is important”, and our benchmark results “extensive”.
>
> > Section 4.1, 4.2 and 4.3 makes a large number of significant assumptions, modifications and improvements to the traditional CE algorithm, it is not clear whether each modification is actually necessary or which are the most important ones.
>
> We are unsure what the reviewer is referring to here, and would love a chance to clear up any misunderstanding in Sections 4.1-4.3.
>
> CCE has three key differences from a traditional CE algorithm: Fusion of matrix-multiplication + cross-entropy loss into a single kernel, gradient filtering, and vocabulary sorting. We ablate all these in Table 1. If the reviewer has requests for specific ablations, we are happy to run them.
>
> Overall, CCE acts as a plug-in replacement for CE with minimal assumptions. We require a linear/classification layer to precede CE, which is true for almost all deep networks we are aware of. To see true gains from CCE this classification layer needs to cover a large number of classes.
>
> > What are the theoretical justifications on why CCE might be much more computationally and memory efficient compared to traditional CE?
>
> CCE is two operations fused together: matrix multiplication followed and cross-entropy loss. The fused kernel executes exactly the same operations as CE, in fact the two have almost identical theoretical FLOPS. However, the fused CCE kernel reduces memory usage by eliminating the need to store intermediary results. CCE improves computation efficiency by exposing more work at once to the GPU.
> The best analogy to this in published literature is FlashAttention, which presents a similar fused kernel for the attention operation. Unlike FlashAttention, CCE does not alter or limit the original operator (CE).
>
> > For example, why can't you apply the same chunking strategies used in CCE for traditional CE?
>
> CCE doesn’t use chunking. CCE uses blocking in service of mapping its computation GPU, but this is distinct from chunking and achieves a different goal.
>
> Chunking strategies can be applied to save memory in the context of traditional CE, as shown by the Liger Kernel and Torch Tune baselines. These come with performance costs and still use considerably more memory than CCE.
>
> It is possible to apply chucking strategies to CCE, but this would not save memory and likely harm performance.
>
> > Preliminaries (section 3) does not adequately prepare the reader for the complexity of the notation in section 4.
>
> As suggested by Reviewer HR95 we have added a section in the Appendix to provide more explanation on our notation.
>
> > How does CCE compare to CE on the CPU, or non-parallel cases? Are the improvements algorithmic or does it need to take advantage of GPU parallelization strategies?
>
> The memory savings of CCE would directly apply to CPU implementations. In fact, if one were to write a sequential CPU implementation of a fused linear + CE operation, something like CCE would naturally emerge. Computational improvements may transfer to a CPU-parallel implementation too. The blocking strategy used for GPU matrix multiplication is also commonly used in parallel CPU matrix-multiplication algorithms for the same reasons: it makes efficient use of the cache hierarchy. Gradient-filtering would transfer to even a non-parallel case as it still enables work to be skipped, but vocabulary sorting would not be needed in a non-parallel case.
>
> We focused on the parallel case using a GPU as non-parallel or CPU-parallel is simply too slow to train modern models.

---

### Official Review · Reviewer_XSRM · 2024-11-04

**Soundness:** 4
**Presentation:** 3
**Contribution:** 4
**Rating:** 10
**Confidence:** 4

**Summary:**

This paper proposes "Cut cross-entropy" (CCE), a method that reduces the memory footprint of computing the cross-entropy loss during LM training dramatically, by never materializing the full matrix of logits/probabilities (which can be huge: batch * sequence_length * vocab_size).  To accomplish this, it realizes that the cross-entropy loss can be broken down into two components: (1) the logit for the correct next token, and (2) the log-sum-exp (log of softmax denominator) --- both of these terms are scalars, and can be computed without materializing the full logit tensor. In particular, (1) is computed via simple vector dot-products (Algorithm 1), while (2) can be computed by accumulating partial sums of the log-sum-exp, without every materializing all elements of the sum at once (Algorithm 2).  For the backward pass (Algorithm 3), the paper proposes two methods --- gradient filtering and vocabulary sorting --- that reduce the backward pass time by skipping gradient computations for blocks of the softmax matrix where all values are < 2^(-12).

Putting all these pieces together, CCE is able to match the speed and quality of existing implementations of the cross-entropy loss, while only requiring a very small percentage of HBM memory (e.g., 1 MB instead of 1GB->24 GB).

**Strengths:**

- Reducing the memory requirement for computing the CE loss in LLMs is a strong contribution, especially as the vocabulary sizes, batch sizes, and sequence lengths of LLMs continue to grow.  This custom kernel could save many people lots of time trying to get around OOM errors during training, and make it easier to train models with larger sequence lengths/batch sizes/vocab sizes.
- The algorithm is clever and elegant, taking inspiration from FlashAttention, which avoids materializing full attention score matrix during attention computation.
- The experiments demonstrate that CCE can reduce training memory requirements without impacting quality/convergence during training, or training speed, relative to strong baselines (e.g., torch.compile).

**Weaknesses:**

- I think the section about the backward pass could be explained more clearly (see my questions below for points of confusion that could be clarified).
- I think there could have been additional experiments to explore how CCE performs relative to baselines as different hyperparameters vary (e.g., relative size of vocabulary vs sequence length vs. hidden dim, sparsity of S, etc.).

**Questions:**

- Are there regimes where CCE is meaningfully slower than the torch.compile method?
- There were a few elements of the backward computation that I think could be explained more clearly:
  - What is the "v" index in the lines 339-341 (Algorithm 3)?
  - Why doesn't recomputing the large $C^T E$ matrix multiplication in the backward pass (Algorithm 3) lead to slow-downs? If I understand correctly, this is because although much extra time is spent on this recomputation, less time is spent on the subsequent matrix multiplications, due to the gradient filtering/vocab sorting? Can you break down more granularly how much time each component of CCE (especially the backward pass) takes, and compare this to the naive implementations, so that it becomes clear what is happening here?
  - Can you explain this paragraph in more detail please: "We implement the second matrix multiplication in the main memory of the GPU, as a blockwise implementation would require storing or synchronizing S..."? Here, is "main memory" HBM?
  - I think there may be mistakes in the backward pass equations at the bottom of page 5 (lines 266-269).  Letting V be vocab size, L be sequence length, and D be hidden dimension, we can see that C is [D,V], E is [D,L], and S is [V,L].  Then for the matrix shapes to be correct, shouldn't it be:
    - $d/dE = C (S \cdot LSE_{grad})$ --- which is a [D,V] * [V,L] multiplication, which gives [D,L], which is the correct shape of E,
    - $d/dC = E (S \cdot LSE_{grad})^T$ --- which is a [D,L] * [L,V] multiplication, which gives [D,V], which is the correct shape of C.
  - Can you include, at least in the appendix, a version of algorithm 3 that also includes the backward pass of the indexed matrix multiplication?
  - NIT: I think it could be clearer to update the notation to be something like the following, to make Algorithms 2 and 3 easier to follow. For example, you could use $V$, $L$, $D$ to denote vocab size, sequence length, and hidden dim, and correspondingly $V_B$, $L_B$, and $D_B$ to denote the dimensions of the blocks, and $B_V = V/V_B$, $B_L = L/L_B$, $B_D = D/D_B$ to denote the number of blocks, and v, l, d to index into these blocks.
- Can you add a discussion around sequence parallelism approaches, which can also reduce logit memory per GPU by splitting logits along sequence dimensions?

---

> ### Author Response · Authors · 2024-11-25
> **Response to XSRM (1/2)**
>
> We thank the reviewer for their review. We are pleased that they found our algorithm “clever and elegant”, that reducing the memory requirement for CE loss is a “strong contribution”, and that our work “could save many people lots of time trying to get around OOM errors.”
>
> > I think there could have been additional experiments to explore how CCE performs relative to baselines as different hyperparameters vary (e.g., relative size of vocabulary vs sequence length vs. hidden dim, sparsity of S, etc.).
>
> Thank you for the suggestion. Since submission we have run additional experiments and included them in the updated appendix. These experiments alter the vocabulary size, hidden dimension, and sparsity of S. The sparsity of S is largely determined by the vocabulary size since the number of non-trivial values is largely constant (it is governed by the data type) and thus models with smaller vocabularies have less sparse S matrices.
>
> The trend is that when the model has a high ratio of vocabulary size to hidden dim (e.g., Gemma 2 (2B) where the ration is 111), CCE is faster than torch.compile. When the model has a low ratio (e.g. Phi 3.5 where the ratio is 10), CCE is slower than torch.compile, but continues to save a considerable amount of memory.
>
> We have also added benchmarking with less tokens. CCE exhibits very similar behavior as Baseline and torch.compile — as there are less tokens, it gets faster. Further, because CCE does not utilize chunking, it does not reach a plateaus as performance becomes bound by kernel launch time, not computation time.
>
> > Are there regimes where CCE is meaningfully slower than the torch.compile method?
>
> Simply put, yes. When the ratio of vocabulary size to hidden sizes becomes small, CCE can be meaningfully slower than torch.compile.
>
> In experiments fine-tuning Phi 3.5 Mini (where CCE has the worst relative performance to torch.compile), CCE only increases total training time by 1-2%. However, in our new experiments pre-training Phi 3.5 Mini, CCE increased total training time by 25% as gradient filtering is able to filter out significantly less blocks in this regime.
>
> > Why doesn't recomputing the large $C^T E$ matrix multiplication in the backward pass (Algorithm 3) lead to slow-downs?
>
> Re-computing $C^\top E$ does lead to slowdowns and CCE would be faster if it didn’t need to re-compute this. We are able to offset this by the amount of time saved elsewhere.
>
> > Can you break down more granularly how much time each component of CCE (especially the backward pass) takes, and compare this to the naive implementations, so that it becomes clear what is happening here?
>
> We have broken down the time spent for CCE and the Baseline implementation in their backward passes for Gemma 2 (2B) and updated the Appendix (see C.2). To summarize here:
>
> CCE spends considerably less time on the cross-entropy loss and softcap portions of the gradient computation. For Baseline, these are very memory intensive operations (there is relatively very little computation done). For CCE, the logits are already in SRAM and we do not write the result of this computation to main memory, saving a significant amount of time.
>
> Coincidentally, CCE spends a very similar amount of time computing the gradient wrt. the embeddings while CCE spends less time computing the gradient wrt. the classifier. This is because the axis we reduce along for the classifier, N, is shorter than the axis for the embeddings, |V|, and thus leads to less contention on global memory.
>
> Compared to Baseline, CCE saves 30 ms on the gradient of the logits wrt. cross-entropy loss, 12 ms on the gradient wrt. softcapping, 5 ms on the gradient wrt. E, and 15 ms on the gradient wrt. C. This saving of 62 ms offsets the time spent re-computing and applying the gradient filter.
>
> Unfortunately the implementations using torch.compile are a black-box to us as any attempt to inject profiling or disable parts of the computation alters the computation graph therefore torch.compile’s ability to fuse kernels.
>
> > Can you include, at least in the appendix, a version of algorithm 3 that also includes the backward pass of the indexed matrix multiplication?
>
> Added as algorithm 4. Let us know if you have any suggestions to make it clearer.
>
> > think it could be clearer to update the notation to be something like the following, to make Algorithms 2 and 3 easier to follow. For example, …
>
> Thank you for the suggestion. We have switched from using M to V to denote indexing and blocking along the vocabulary dimension. We chose to continue to use N to denote indexing along the batch/input dimension as L is often used to represent the length of sequences (like the input to self-attention) and we did not want to cause any possible confusion as to if CCE has temporal dependencies (it does not).

---

> ### Author Response · Authors · 2024-11-25
> **Response to XSRM (2/2)**
>
> > I think there may be mistakes in the backward pass equations at the bottom of page 5 (lines 266-269)…
>
> Thank you for catching this, we have updated the paper to correct for this dimension mismatch.
>
> > Can you explain this paragraph in more detail please: "We implement the second matrix multiplication in the main memory of the GPU, as a blockwise implementation would require storing or synchronizing S…”? Here, is "main memory" HBM?
>
> First, let us clarify that the terms HBM and main memory of often used interchangeably and refer to the same thing here. We use main memory as HBM refers to a specific memory technology that used in AI-focused GPUs (e.g., A100 and H100), but other technologies (e.g., GDDR6 and GDDR6x) may also fulfill the role of main memory.
>
> Now, on to the paragraph in question. In any matrix multiplication, there are the two outer dimensions and the inner dimensions that is reduced over. In a canonical GPU matrix multiplication kernel, the reduction along the inner dimension is performed in SRAM (like the $D$ dimension in Algorithm 2, L279). This memory is extremely fast, but local to a specific warp or block.
>
> For the $\nabla E^\top = S C$ and $\nabla C^\top = S^\top E$ matrix multiplications, the reduction of the inner dimension, $V$ and $N$, respectively, is performed in GPU main memory.
>
> We do this because of the re-computation of the logits, $C^\top E$. To compute the gradient, we must first recompute the logits and then use them to compute S. Here we follow the canonical GPU matrix kernel and perform the reduction along the $D$ dimension in SRAM. Turning to the two remaining matrix multiplications, the full matrix S has been block-divided amongst all the different CUDA blocks and thus no single block has all the values needed to reduce the inner dimension in SRAM. From here, there are two options: 1) synchronization and storage of S in main memory (which will eliminate our memory savings) or 2) perform the reduction in main memory (which has a performance cost due to the relatively slower memory). We choose the latter and developed gradient filtering to offset the performance cost.
>
> > Can you add a discussion around sequence parallelism approaches, which can also reduce logit memory per GPU by splitting logits along sequence dimensions
>
> Great idea! Done in updated PDF.

---

> > ### Comment · Reviewer_XSRM · 2024-12-02
> >
> > I thank the authors for their thorough responses to my questions. I leave my score unchanged.

---

### Meta-Review · Area_Chair_KC3K · 2024-12-16

**Metareview:**

This paper introduces a method called "Cut Cross-Entropy" (CCE) to reduce the memory consumption of the cross-entropy layer in LLMs. As the vocabulary size grows and various memory optimization techniques are applied to LLMs, memory consumption increasingly shifts from weights and activations to the cross-entropy layer. This paper presents several strategies to mitigate the memory usage of this layer and achieve significant memory savings.

Main strengths:
- Very novel insight
- Significantly improved performance

Main weaknesses:
- As several reviewers point out, the paper's clarity could be improved
- Lack of substantial studies on different architectures, hyperparameters, etc.

**Additional Comments On Reviewer Discussion:**

During the rebuttal, the authors clarified lots of questions. They also provided several additional experiments: small-scale pre-training on various architectures, ablations on vocabulary size, hidden dimension, etc.,

---

### Decision · Program_Chairs · 2025-01-22

Accept (Oral)